# HUMAN-IN-THE-LOOP TARGETED MOLECULE DESIGN INFORMED BY TRANSCRIPTOMES

## ABSTRACT

Transcriptome-guided targeted molecule design contributes to the advancement of precision medicine. However, the fragmented and poorly integrated nature of existing datasets limits the ability of transcriptome-aware approaches to meet the design expectations of chemists, often requiring additional expert intervention after molecular generation. To address this gap, we propose HITGEN, a **h**uman-**i**n-the-loop **t**argeted molecule **gen**eration framework informed by transcriptomes. Specifically, HITGEN operates in two phases: i) Transcriptomes are served as the central biological driver and are fused with expert a posteriori knowledge via a tailored bidirectional attention mechanism, enabling biochemically grounded guidance for diffusion-based generation of molecule candidates. ii) An expert-guided human-in-the-loop optimization mechanism is employed by HITGEN to refine these candidates toward desired molecular targets. Extensive experiments demonstrate that HITGEN consistently outperforms state-of-the-art models across ten evaluation metrics, yielding chemist-aligned targeted molecules with potential for precision medicine. The code will be released upon acceptance of the paper.

## 1 INTRODUCTION

The transcriptome, defined as the full complement of RNA transcripts expressed in a cell under specific physiological or pathological conditions, offers a dynamic molecular snapshot of cellular function and biological activity (Liu et al., 2024). Owing to their biological richness and chemical informativeness, transcriptomes serve as a critical resource for understanding specific disease mechanisms, identifying therapeutic targets, and characterizing complex biological contexts (Namba et al., 2025). Recent advances in artificial intelligence have further deepened the exploration of transcriptome data, integrating it into multiple stages of drug discovery and particularly accelerating targeted molecule design (Li & Yamanishi, 2025a; Cheng et al., 2024). Generating molecules conditioned on transcriptomes has shown great promise in enhancing molecular responsiveness to biological states and improving therapeutic relevance.

Despite substantial progress, existing methods remain constrained by the limited practical flexibility in customizing transcriptomic conditions for real-world discovery pipelines. This limitation restricts truly interactive feedback between researchers and models, especially in targeted molecule design scenarios that demand strict biochemical constraints and tight experimental control (Qi et al., 2024; Wagle et al., 2024). A key contributing factor is the inherent barrier across datasets, which makes it difficult to align or transfer chemical information across diverse transcriptomic contexts and experimental platforms (Liu & Jin, 2025). Meanwhile, single-source datasets often fail to provide sufficient auxiliary information and contextual signals to enhance the interpretation of transcriptomes or the quality of generated outputs (Baysoy et al., 2023). Moreover, the abstract and high-dimensional nature of transcriptomic inputs often lacks explicit, human-readable semantic structure, further exacerbating the difficulty of representation learning (Yang et al., 2025). This poses a substantial obstacle for generative models to reliably extract biologically meaningful information. These challenges suggest the need for an expert-guided, participatory representation that serves as an indirect interpretation of the transcriptome, thereby deepening biochemical understanding of the cellular environment and enabling more precisely and reliably tailored molecule generation (Luo et al., 2024).

A promising direction lies in incorporating molecular descriptions (e.g., desired biochemical properties, design priorities) to semantically and contextually guide transcriptome-aware molecule design.

Given the proven potential of natural language in molecule design, leveraging such chemical text guidance to robustly and adaptively regulate design conditioned on transcriptome may provide a more accessible and biologically consistent interface for conditional molecule design, particularly in designing target-specific molecules with enhanced biological compatibility and chemical explainability (Edwards et al., 2021; Gong et al., 2024). This direction also introduces several key technical and conceptual challenges that merit deeper investigation. First, existing studies on molecule design rarely address how to effectively integrate the transcriptomic data, leading to a lack of technically cohesive generative frameworks (Jiang et al., 2025). In addition, the textual descriptions used to guide transcriptome-aware targeted molecule design differ fundamentally from conventional natural language inputs typically used in generative models. Instead of representing transcriptome profiles or molecular structures, these descriptions provide chemical-level expectations about the molecular consequences of transcriptomic expression (Shi et al., 2025). This paradigm imposes multiple constraints on molecule design, such as preserving biological context and incorporating chemical semantics. Such complexity highlights the necessity of developing well-structured feature learning and integration strategies at the algorithmic level.

As illustrated in Figure 1, we address the aforementioned challenges by introducing expert guidance from human-in-the-loop perspective to guide transcriptome-aware targeted molecule design. We propose a next-generation framework for **h**uman-**i**n-the-loop **t**argeted molecule **gen**eration (HiTGen). Specifically, HiTGen facilitates multi-perspective transformation and integration of transcriptome profiles through the introduction of human-in-the-loop guidance, enhanc-

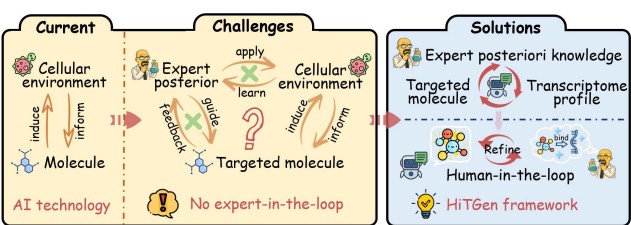

Figure 1: HiTGen uses an expert-guided human-in-the-loop process to design molecules, addressing the fundamental challenges in transcriptome-aware targeted molecule design.

ing both the biological consistency and chemical interpretability of the designed targeted molecules. Molecule-induced transcriptome profiles serve as the biological information, capturing the human cell environment, while the expert-guided design descriptions, derived from the molecules, represent the chemical information, reflecting intended chemical objectives. HiTGen then applies bidirectional attention and feature fusion to the information representations, guiding the conditional diffusion model to generate targeted molecules that are both biologically matched and chemically analogous with the expectations. Furthermore, we devise a human-in-the-loop optimization mechanism that supports iterative refinement of candidate molecules, ultimately yielding high-quality molecules for downstream new drug development. The main contributions are as follows:

- **Expert-guided transcriptome-aware molecule design framework**: HiTGen introduces an expert-guided perspective that integrates transcriptome profiles with expert a posteriori knowledge to generate targeted molecules with both biological relevance and chemical alignment.

- **Human-in-the-loop molecule optimization**: A highly scalable human-in-the-loop optimization mechanism improves the therapeutic efficacy of the designed targeted therapeutics and enables seamless deployment within drug-discovery workflows.

- **Outperforming state-of-the-art (SOTA) methods**: HiTGen achieves consistent improvements over SOTA methods across diverse evaluation metrics, demonstrating its effectiveness and generalizability in targeted molecule design and optimization.

## 2 RELATED WORK

**Transcriptomic biological applications.** Transcriptome profiles, which capture the expression landscape under specific physiological or pathological conditions, have been widely used in various areas of biomedical research, such as disease subtype classification, mechanism-of-action inference, and drug discovery (Fukushima-Nomura et al., 2025; Tong et al., 2024; Jose et al., 2024). Recent progress in artificial intelligence has further accelerated research in these domains by enhancing the ability to extract meaningful signals from transcriptomic datasets (Li & Yamanishi, 2025b). Despite these successes, existing studies predominantly treat RNA-level data as auxiliary features

for predictive tasks (Qi et al., 2024). In particular, during the target molecule design phase of drug discovery, expression information is rarely utilized as a central generative condition (Shayakhmetov et al., 2020). Therefore, research on designing molecules directly conditioned on transcriptome profiles remains limited, revealing a critical gap in current generative modeling efforts.

**Challenges of targeted molecule design.** A major barrier to applying transcriptome in molecule design lies in the difficulty of mapping transcriptomic expressions to molecular structures. For instance, early studies employed molecules known to induce specific transcriptomic shifts as inputs to generative adversarial networks for de novo molecule design (Méndez-Lucio et al., 2020). In such settings, the transcriptome was represented only indirectly through the inducing molecules, rather than being explicitly modeled. Instead, TRIOMPHE (Kaitoh & Yamanishi, 2021) utilized induced expression patterns to retrieve biologically similar molecules as priors for design, but failed to position transcriptomes as the core generative driver. More recent efforts, such as GxRNN (Matsukiyo et al., 2024), directly utilized transcriptome as a conditional input for recurrent neural networks, offering a biologically aligned approach to target molecule design. Notably, these profiles often exhibit high sparsity and substantial noise following chemical perturbation, which undermines both the robustness and interpretability of the generative process. GxVAEs (Li & Yamanishi, 2025a), the current SOTA model, and its extension HNN2Mol (Li & Yamanishi, 2025b) addressed this limitation by leveraging molecular perturbations to biologically anchor transcriptomic inputs and applying variational autoencoders (VAE) (Jin et al., 2018) to mitigate data sparsity. Nevertheless, these approaches rely on a single conditioning strategy and still fall short in capturing the full biological context encoded in transcriptome. This underscores the urgent need for deeper and more structured interpretations of transcriptome profiles to support precise and controllable targeted molecule design.

**Text description-guided molecule generation.** A persistent gap in transcriptome-conditioned molecule design lies in the lack of auxiliary information that enables deeper interpretation of biochemical context (Baysoy et al., 2023). In particular, inherent barriers across datasets make it challenging to integrate or align transcriptome across domains (Yu et al., 2023; Liu & Jin, 2025). As a result, multi-perspective analysis of transcriptomic expression for guiding target molecule design remains largely unexplored. Recent advances in leveraging natural language for molecular modeling have exposed the potential of textual information to guide molecule design. For example, Text2Mol (Edwards et al., 2021) introduced a BERT-based (Devlin et al., 2019) framework that aligns chemical structures with semantic descriptions through interpretable association rules and a shared embedding space. MolT5 (Edwards et al., 2022) and Text+ChemT5 (Christofidellis et al., 2023) further leveraged the language modeling capabilities of the T5 architecture to enable bidirectional translation between molecules and chemical texts. More recently, TGM-DLM (Gong et al., 2024) addressed the autoregressive limitations of prior models by adopting a diffusion-based framework for controllable text-conditioned molecule design. However, TGM-DLM, despite being the current SOTA model, underutilizes the rich semantic guidance of textual inputs in its second-phase correction, which solely focuses on ensuring validity without enhancing overall generation quality. These studies collectively underscore the promise of natural language as an effective and expressive modality for molecule design. This insight suggests a new direction: by incorporating targeted molecular descriptions as semantic complements to transcriptome-derived representations, designers may establish richer biochemical associations across streams, effectively bridging transcriptomic biological expression with intended chemical properties to enable precise and tailored molecule design.

In this study, we first derive human-in-the-loop design descriptions based on perturbation molecules associated with the transcriptome. These textual descriptions are crafted to regulate the binding abilities of designed molecules in accordance with the chemist's intended objectives. HiTGen subsequently performs a multi-perspective integration of biochemical information and conditions the diffusion model accordingly to guide targeted molecule design. Notably, HiTGen incorporates a human-in-the-loop optimization mechanism, which ensures the production of high-quality molecules while simultaneously yielding a diverse set of promising candidates for downstream drug discovery.

## 3 HiTGen

Figure 2 illustrates the architecture of HiTGen. The expert-assisted guidance step supplements the input profiles in a human-in-the-loop manner, while the information integration component learns and

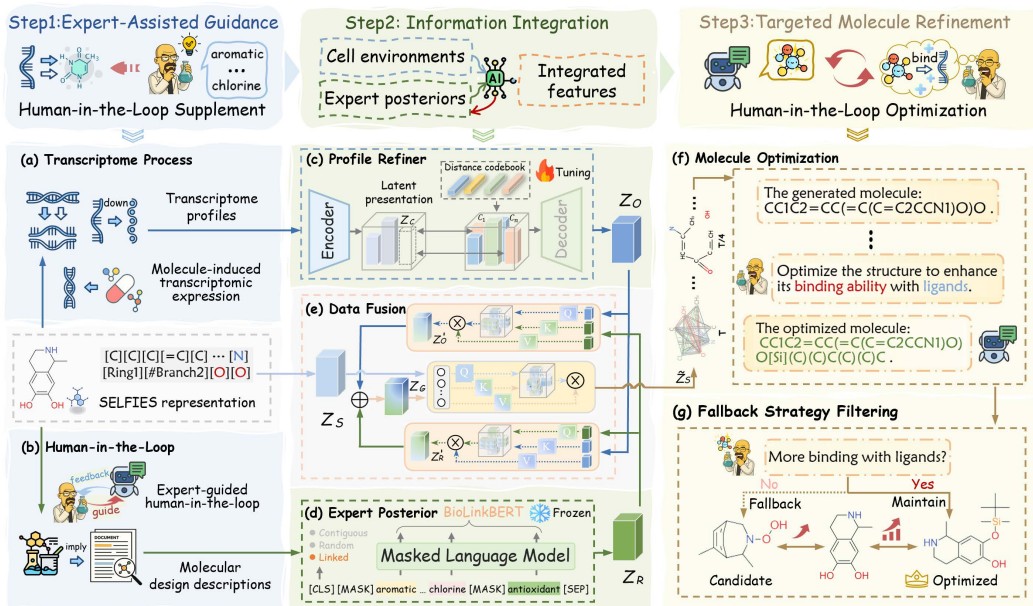

Figure 2: Overview: **Step 1: expert-assisted guidance** supplements (a) the drug-induced transcriptome process via molecular descriptions, derived through (b) an expert-guided human-in-the-loop manner that incorporates expert a posteriori knowledge. **Step 2: information integration** utilizes (c) the profile refiner and (d) the expert posterior to collaboratively generate $\mathbf{Z}_O$ and $\mathbf{Z}_R$. Then, (e) the data fusion module applies tailored bidirectional cross-attention and multi-source feature fusion to these representations, producing a unified representation $\mathbf{Z}_G$, which conditions the diffusion decoder to guide the generation of $\mathbf{Z}_S$. **Step 3: targeted molecule refinement** generates a batch of candidate targeted molecules using the diffusion model, followed by (f) human-in-the-loop molecule optimization and (g) fallback strategy filtering to robustly enhance their overall design quality.

regulates multi-dimensional features from the transcriptome. Finally, a targeted molecular refinement paradigm is introduced to design candidate molecules and enhance them through biochemically grounded human-in-the-loop optimization.

## 3.1 EXPERT-ASSISTED GUIDANCE

A transcriptome profile $E = [e_1, e_2, \ldots, e_K]$, where $e_k$ denotes the $k$-th expression level induced by a known drug and $K$ is the dimensionality of the profile, faithfully reflects the underlying chemical mechanism of the drug responsible for the observed biological perturbation. To interpret these mechanisms, we first obtain the self-referencing embedded strings (SELFIES) (Krenn et al., 2020) representation of the known molecule, which serves as a robust bridge between molecular structure and chemical semantics. SELFIES is chosen for its syntactic regularity and intrinsic validity, ensuring reliable chemical structure encoding for language-based description. We then employ BioT5 (Pei et al., 2023) as the expert chemist–supervised model to generate molecular descriptions under close chemist supervision, thereby supplementing the transcriptome with design information. This process transforms the implicit biochemical cues in the transcriptome into semantically controllable formats for designers. Notably, with the molecular representation as a shared anchor, the transcriptome and molecule description form a one-to-one correspondence, reinforcing the association between structure, biological context, and chemical semantics.

## 3.2 INFORMATION INTEGRATION

Given a transcriptomic matrix $\mathbf{O} \in \mathbb{R}^{M \times K}$, where $M$ is the number of samples, we first encode each input into a robust latent representation $\mathbf{Z}_e \in \mathbb{R}^{M \times D}$ using a multi-layer perceptron with $\tanh$ activations and dropout layers to better mitigate overfitting, where $D$ is the dimension of $\mathbf{Z}_e$. The encoded features $\mathbf{Z}_e$ are reshaped into $\mathbf{Z}_C \in \mathbb{R}^{M \times D \times H}$ to simulate a pseudo multi-token structure, where $H$ is a reshaped feature dimension used for vector quantization. These features are then

projected onto a learnable, finite codebook $\mathcal{C} = \mathbf{c}_1, \ldots, \mathbf{c}_N$, where each codebook entry $\mathbf{c}_n \in \mathbb{R}^D$ and $N$ denotes the number of discrete codes. Using efficient nearest neighbor search, each $\mathbf{Z}_C$ is matched to its closest code, resulting in a quantized latent vector $\mathbf{Z}_O \in \mathbb{R}^{M \times D}$.

The quantization process introduces a loss term $\mathcal{L}_O$, which penalizes the deviation between the encoder output and its closest codebook vector (Van den Oord et al., 2017). A decoder composed of symmetric linear layers with $\tanh$ activations maps $\mathbf{Z}_O$ back to the original feature expression space, yielding a reconstructed output. The model is trained to minimize:

$$\mathcal{L}_O = \|f(\mathbf{Z}_O) - \mathbf{O}\|_2^2 + \|\text{sg}(\mathbf{Z}_e) - \mathbf{Z}_O\|_2^2 + \beta \|\mathbf{Z}_e - \text{sg}(\mathbf{Z}_O)\|_2^2, \tag{1}$$

where $f(\cdot)$ denotes the decoder function and $\text{sg}(\cdot)$ is the stop-gradient operator that prevents updates. The hyperparameter $\beta$ controls the strength of the commitment penalty.

To complement the transcriptomic profile $\mathbf{O}$ with semantic-level expert guidance, we define a semantic embedding objective aimed at capturing chemically meaningful patterns from molecule design descriptions. Given a natural language description $F$ that reflects the chemical specification associated with the $\mathbf{O}$, we process $R$ using BioLinkBERT (Yasunaga et al., 2022), a domain-specific pre-trained language model. The input text $R$ is first tokenized and normalized to a maximum length $L$, where only chemically relevant tokens such as "benzene" and "ether" are preserved. In this process, [CLS] indicates the beginning of the sequence, [MASK] is applied to mask tokens lacking chemical relevance, and [SEP] functions as a delimiter between segments. The resulting sequence is formatted as [CLS] $R_1$ [MASK] $R_2$ [SEP], where $R_1$ and $R_2$ denote two chemical segments. The final hidden state corresponding to the [CLS] token is extracted as the semantic embedding $\mathbf{Z}_R \in \mathbb{R}^D$, serving as a high-level representation of the input text. The model is pre-trained using a hybrid objective that combines masked language modeling and domain-relation prediction, formulated as:

$$\mathcal{L}_R = -\sum_i \log p(r_i \mid \mathbf{h}_i) - \log p(b \mid \mathbf{h}_{\text{[CLS]}}), \tag{2}$$

where $r_i$ is each token in the sequence, $\mathbf{h}_i$ is contextualized representation, $i$ corresponds to the range of $L$, and $b$ represents the relation type between segments encoded.

We fuse transcriptomes and descriptions to jointly guide the design of biochemically aligned molecules. This part integrates the transcriptome-derived representation $\mathbf{Z}_O$ and the chemical semantic embedding $\mathbf{Z}_R$ into a shared, unified latent space. The motivation behind this design lies in bridging the persistent semantic gap between transcriptome profiles and chemical understanding, two highly heterogeneous yet biologically intertwined information sources. This module explicitly models reciprocal, bi-directional attention between information to allow each stream to contextually recalibrate itself with respect to the other. This fosters a dynamic, fine-grained feature refinement process, which is essential for capturing subtle dependencies. Specifically, we employ two parallel cross-attention layers (Chen et al., 2021) and the $\mathbf{Z}_O$ attends to the $\mathbf{Z}_R$:

$$\mathbf{Z}_O' = \mathbf{Z}_O + \text{CA}(\mathbf{Z}_O, \mathbf{Z}_R, \mathbf{Z}_R), \tag{3}$$

where $\mathbf{Z}_O$ serves as the query while $\mathbf{Z}_R$ is used for keys and values. This process injects chemical semantics back into the biological representation, enabling the transcriptomic vector to be aware of its interpreted textual meaning. Similarly, $\mathbf{Z}_D$ updates through cross-attention over the transcriptome:

$$\mathbf{Z}_R' = \mathbf{Z}_R + \text{CA}(\mathbf{Z}_R, \mathbf{Z}_O, \mathbf{Z}_O). \tag{4}$$

This reverse attention enables each chemical token to capture nuanced, condition-aware refinements grounded in transcriptomic signals, thereby aligning local chemical semantics with the global biological context and improving the fidelity of downstream molecule generation.

After the bidirectional updates, we concatenate the refined embeddings and apply a fusion layer to normalize the joint space: $\mathbf{Z}_G = \text{LN}(\mathbf{Z}_R' \oplus \mathbf{Z}_O')$, where $\oplus$ denotes concatenation along the feature dimension, and $\text{LN}(\cdot)$ indicates normalization. The fused embedding $\mathbf{Z}_G$ then conditions the Transformer-based diffusion decoder, guiding the molecular embedding $\mathbf{Z}_S$ via embedded cross-attention to obtain the molecular hidden state $\tilde{\mathbf{Z}}_S$, which retains biologically grounded transcriptomic context and enhances chemical interpretability.

### 3.3 TARGETED MOLECULE REFINEMENT

In this step, the design and refinement of molecules are performed in two tightly coupled successive stages. In the first stage, $\tilde{\mathbf{Z}}_S$ is processed by the conditional diffusion model via a decoder to yield

the predicted token embedding matrix $\mathbf{X}_0$. Each column vector $\mathbf{x}_i$ in $\mathbf{X}_0$ is then mapped to its nearest-neighbor token in the vocabulary embedding space $\mathcal{V}$ using the $L_2$ distance, thereby faithfully reconstructing the discrete molecular sequence.

$$\hat{w}i = \arg\min w_j \in \mathcal{V} \left\| \mathbf{x}i - \mathbf{g}(w_j) \right\|_2 , \tag{5}$$

where $\mathbf{g}(\cdot)$ is the embedding of token $w_j$. This rounding operation maps continuous diffusion outputs back to a coherent molecular sequence. All generated molecules at this stage are treated as candidates for subsequent interactive optimization.

With candidate molecules in place, we construct a human-in-the-loop architecture based on DrugAssist to guide the second-stage molecule optimization (Ye et al., 2025). During the prompting, a reference drug molecule is provided, and the model is instructed to optimize the candidates with respect to both pharmacological properties and binding affinity, aiming to enhance their similarity to the known drug. Prompts are detailed in Appendix C. In this interaction-driven setting, experts are able to refine molecular designs based on their intended objectives. Concurrently, we incorporate a fallback strategy: if the optimized molecule exhibits lower quality compared to its first-stage counterpart, the original candidate is retained as the final output. This mechanism addresses two critical issues: i) over-optimization that causes the designed molecule to deviate from the biological expectations encoded in the transcriptome; ii) negative optimization that yields suboptimal molecules or chemical structures that are misaligned with the expert's intended design objectives.

## 4 EXPERIMENTS

### 4.1 DATASETS AND METRICS

**Datasets.** The dataset comprises three components: *transcriptomic inductions*, which consist of transcriptome profiles from human breast cancer cell line under molecular induction (Duan et al., 2014), paired with the corresponding molecules in SELFIES; *transcriptomic perturbations*, which include transcriptomic expressions from 77 human cell lines subjected to chemical perturbations, covering eight overexpression signatures (AKT1, AKT2, AURKB, CTSK, EGFR, HDAC1, MTOR, and PIK3CA) and two knockdown signatures (SMAD3 and TP53) associated with cancer-related targets; and *morbid transcriptomes*, which capture disease-specific transcriptomic signatures from patients with gastric cancer and Alzheimer's disease (Wang et al., 2016), where multiple individual profiles were averaged to represent each condition. Notably, transcriptomic inductions serve as the training set, while transcriptomic perturbations and morbid transcriptomes are utilized for validation and case study. Further details about datasets are provided in Appendix B.

**Metrics.** The following seven indicators were adopted to assess the overall quality of designed targeted molecules. *Total* is a composite score that jointly considers validity, uniqueness, and novelty. *Leven* (Levenshtein) measures the minimum transform distance between molecular string and normalizes it to evaluate their similarity. *Fréchet chemnet distance* (FCD) quantifies distributional similarity by comparing the latent representations of designed molecules and reference molecules. *Overlap* assesses the extent to which functional groups are preserved between the designed and original molecules. *MACCS*, *RDK*, and *Morgan* scores compute the average Tanimoto similarity between the designed and ground-truth fingerprints from three hierarchical perspectives: molecular structure, atom paths, and individual atoms, respectively. Details of these metrics are in Appendix E.

### 4.2 HITGEN WITH HUMAN-IN-THE-LOOP ADVANTAGE

Table 1 presents the comparative results between HITGEN and the current SOTA models that leverage either transcriptomes or textual descriptions. Each result is reported as the mean $\pm$ standard deviation over five runs. Benefiting from the supplementary chemical semantics derived from human-in-the-loop and the incorporation of the advanced framework, HITGEN consistently achieves superior and stable results across all evaluation metrics. With respect to transcriptomic profile reconstruction, Figure H.4 showcases that the HITGEN decoder accurately reconstructs transcriptomic profiles, closely aligning with the original distributions. Metrics such as Total, Leven, and FCD indicate that the designed molecules exhibit significant improvements at the character level after interactive linguistic refinement, for instance, Total improves from 0.71 to 0.98. In terms of structural similarity, HITGEN delivers impressive gains in Tanimoto scores across multiple granularity levels, from coarse-grained MACCS and RDK to the fine-grained Morgan fingerprint. Notably, the Morgan similarity,

Table 1: Comparison of HiTGen with SOTA baseline methods across all evaluation metrics. Bolded values in the blue box indicate the best results.

| | Total ↑ | Validity (%)↑ | Unique (%)↑ | Novelty (%)↑ | Leven ↓ | FCD ↓ | MACCS ↑ | RDK ↑ | Morgan ↑ |
|---|---|---|---|---|---|---|---|---|---|
| TRIOMPHE | 0.35 ± 0.02 | 50.75 ± 0.02 | 78.68 ± 0.01 | 87.57 ± 0.01 | 0.67 ± 0.01 | 0.55 ± 0.01 | 0.39 ± 0.01 | 0.31 ± 0.01 | 0.17 ± 0.01 |
| GxRNN | 0.50 ± 0.01 | 79.23 ± 0.01 | 79.95 ± 0.01 | 78.77 ± 0.01 | 0.67 ± 0.01 | 0.52 ± 0.01 | 0.45 ± 0.02 | 0.34 ± 0.01 | 0.20 ± 0.01 |
| HNN2Mol | 0.55 ± 0.01 | 87.80 ± 0.01 | 81.42 ± 0.01 | 76.31 ± 0.01 | 0.62 ± 0.01 | 0.52 ± 0.01 | 0.47 ± 0.01 | 0.36 ± 0.01 | 0.19 ± 0.01 |
| GxVAEs | 0.67 ± 0.01 | 87.83 ± 0.01 | 86.57 ± 0.01 | 88.95 ± 0.01 | 0.60 ± 0.01 | 0.51 ± 0.01 | 0.49 ± 0.01 | 0.36 ± 0.01 | 0.22 ± 0.01 |
| Mol-T5 | 0.71 ± 0.02 | 77.88 ± 0.02 | 93.21 ± 0.02 | 97.66 ± 0.01 | 0.79 ± 0.01 | 0.17 ± 0.01 | 0.54 ± 0.02 | 0.34 ± 0.01 | 0.21 ± 0.01 |
| Text+ChemT5 | 0.63 ± 0.02 | 67.12 ± 0.01 | 96.01 ± 0.02 | 97.78 ± 0.01 | 0.84 ± 0.01 | 0.21 ± 0.01 | 0.58 ± 0.01 | 0.35 ± 0.01 | 0.20 ± 0.01 |
| TGM-DLM | 0.71 ± 0.01 | 82.33 ± 0.01 | 93.74 ± 0.01 | 91.73 ± 0.01 | 0.52 ± 0.01 | 0.17 ± 0.01 | 0.55 ± 0.01 | 0.37 ± 0.01 | 0.23 ± 0.01 |
| HiTGen | **0.98 ± 0.01** | **100.0 ± 0.00** | **99.71 ± 0.01** | **98.22 ± 0.01** | **0.45 ± 0.01** | **0.17 ± 0.01** | **0.63 ± 0.01** | **0.51 ± 0.01** | **0.40 ± 0.01** |

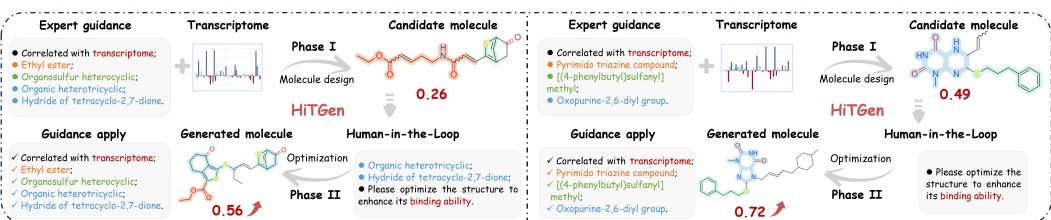

Figure 3: HiTGen aligns expert intent with designed molecules: higher similarity and binding ability.

which most directly reflects structural alignment, showcases a 74% improvement over the best baseline (0.23 to 0.40). Further results in Table G.2 indicate that the integrated refinement module surpasses existing approaches, underscoring the overall effectiveness of HiTGen.

Figure 3 illustrates the advanced advantages of HiTGen, which combines expert guidance with a human-in-the-loop manner to design molecules that adhere to scaffold-level expert intent and are further refined by iterative feedback. This process raises Tanimoto similarity from 0.26 to 0.56 and from 0.49 to 0.72 (115% and 47% increases, respectively), with parallel improvements in binding affinity. Beyond overall molecular quality, we further evaluate the pharmacological fidelity of designed molecules through functional group Overlap analysis. As functional groups often underlie a molecule's biological activity, their retention serves as a proxy for biological relevance. Figure G.2 visualizes the functional group distribution within the tran-

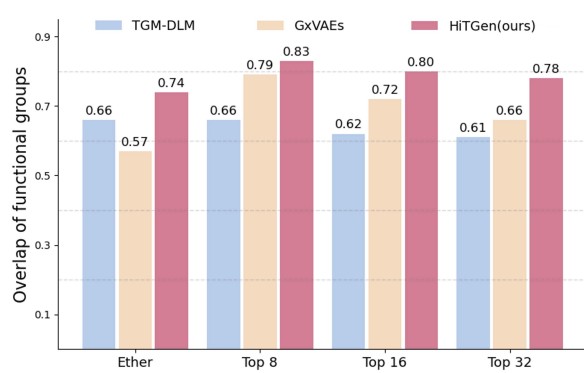

Figure 4: Comparison of structure-based functional group overlap scores between HiTGen and other SOTA models across different group settings.

scriptomic inductions. We then examine functional group Overlap at both the molecular and textual levels, comparing HiTGen with SOTA models based on transcriptomic and textual information. As illustrated in Figure 4 and Figure G.3, HiTGen consistently secures the highest functional group Overlap scores across all evaluation settings. A margin of up to 0.12 over the strongest competitors reveals the robustness and biochemical alignment of our design paradigm.

## 4.3 EVALUATION RESULTS OF HiTGen-DESIGNED MOLECULES

Target-specific ligand design requires molecular structures to be well aligned with the intended protein targets. This study assesses the ability of HiTGen in designing ligand candidates from the transcriptomic perturbations dataset. As shown in Table 2, HiTGen consistently outperforms all single-information SOTA approaches across ten target-specific molecule design tasks. This strong performance can be attributed to the synergistic integration of two molecular string formats and the carefully crafted fallback strategy, enabling HiTGen to achieve statistically significant gains. Note that Total metric exhibits a maximum improvement of 0.52 over the best baseline, emphasizing substantial enhancements in validity, uniqueness, and novelty of the designed molecules. The

Table 2: Comparative evaluation results of HITGEN and current SOTA generative methods on targeted molecule design across ten target proteins.

| Metrics | Methods | AKT1 | AKT2 | AURKB | CTSK | EGFR | HDAC1 | MTOR | PIK3CA | SMAD3 | TP53 |
|---|---|---|---|---|---|---|---|---|---|---|---|
| Total ↑ | GxVAEs | $0.78\pm0.02$ | $0.80\pm0.02$ | $0.80\pm0.02$ | $0.85\pm0.02$ | $0.83\pm0.02$ | $0.73\pm0.03$ | $0.84\pm0.02$ | $0.84\pm0.02$ | $0.78\pm0.02$ | $0.81\pm0.02$ |
| | TGM-DLM | $0.59\pm0.05$ | $0.65\pm0.05$ | $0.67\pm0.04$ | $0.64\pm0.06$ | $0.66\pm0.07$ | $0.61\pm0.05$ | $0.70\pm0.02$ | $0.62\pm0.02$ | $0.57\pm0.04$ | $0.46\pm0.05$ |
| | HITGEN | $\mathbf{0.96\pm0.02}$ | $\mathbf{0.98\pm0.02}$ | $\mathbf{0.97\pm0.02}$ | $\mathbf{0.98\pm0.02}$ | $\mathbf{0.98\pm0.01}$ | $\mathbf{0.98\pm0.01}$ | $\mathbf{0.99\pm0.01}$ | $\mathbf{0.98\pm0.02}$ | $\mathbf{0.99\pm0.01}$ | $\mathbf{0.98\pm0.02}$ |
| Leven ↓ | GxVAEs | $0.89\pm0.02$ | $0.89\pm0.02$ | $0.83\pm0.03$ | $0.88\pm0.02$ | $0.74\pm0.02$ | $0.89\pm0.02$ | $0.78\pm0.01$ | $0.82\pm0.02$ | $0.72\pm0.02$ | $0.77\pm0.02$ |
| | TGM-DLM | $0.88\pm0.02$ | $0.83\pm0.02$ | $0.83\pm0.01$ | $0.94\pm0.02$ | $0.76\pm0.03$ | $0.84\pm0.01$ | $0.82\pm0.02$ | $0.84\pm0.02$ | $0.70\pm0.01$ | $0.71\pm0.01$ |
| | HITGEN | $\mathbf{0.68\pm0.03}$ | $\mathbf{0.65\pm0.02}$ | $\mathbf{0.67\pm0.02}$ | $\mathbf{0.76\pm0.02}$ | $\mathbf{0.68\pm0.02}$ | $\mathbf{0.68\pm0.03}$ | $\mathbf{0.73\pm0.01}$ | $\mathbf{0.75\pm0.03}$ | $\mathbf{0.64\pm0.02}$ | $\mathbf{0.62\pm0.01}$ |
| FCD ↓ | GxVAEs | $0.75\pm0.02$ | $0.78\pm0.01$ | $0.80\pm0.02$ | $0.74\pm0.01$ | $0.83\pm0.01$ | $0.74\pm0.01$ | $0.72\pm0.01$ | $0.91\pm0.01$ | $0.67\pm0.01$ | $0.72\pm0.01$ |
| | TGM-DLM | $0.71\pm0.03$ | $0.73\pm0.02$ | $0.70\pm0.03$ | $0.65\pm0.04$ | $0.70\pm0.03$ | $0.70\pm0.02$ | $0.80\pm0.02$ | $0.62\pm0.00$ | $0.67\pm0.02$ | $0.75\pm0.01$ |
| | HITGEN | $\mathbf{0.32\pm0.01}$ | $\mathbf{0.32\pm0.01}$ | $\mathbf{0.31\pm0.01}$ | $\mathbf{0.24\pm0.01}$ | $\mathbf{0.34\pm0.01}$ | $\mathbf{0.35\pm0.02}$ | $\mathbf{0.41\pm0.01}$ | $\mathbf{0.45\pm0.01}$ | $\mathbf{0.22\pm0.01}$ | $\mathbf{0.31\pm0.01}$ |
| Overlap ↑ | GxVAEs | $0.51\pm0.01$ | $0.46\pm0.01$ | $0.51\pm0.01$ | $0.47\pm0.01$ | $0.44\pm0.01$ | $0.49\pm0.02$ | $0.50\pm0.01$ | $0.51\pm0.01$ | $0.51\pm0.01$ | $0.55\pm0.01$ |
| | TGM-DLM | $0.46\pm0.01$ | $0.47\pm0.01$ | $0.50\pm0.01$ | $0.45\pm0.01$ | $0.42\pm0.01$ | $0.46\pm0.01$ | $0.48\pm0.01$ | $0.53\pm0.01$ | $0.49\pm0.01$ | $0.53\pm0.01$ |
| | HITGEN | $\mathbf{0.61\pm0.01}$ | $\mathbf{0.60\pm0.01}$ | $\mathbf{0.62\pm0.01}$ | $\mathbf{0.64\pm0.01}$ | $\mathbf{0.63\pm0.01}$ | $\mathbf{0.63\pm0.01}$ | $\mathbf{0.59\pm0.01}$ | $\mathbf{0.56\pm0.01}$ | $\mathbf{0.58\pm0.01}$ | $\mathbf{0.66\pm0.01}$ |
| MACCS ↑ | GxVAEs | $0.39\pm0.01$ | $0.39\pm0.01$ | $0.40\pm0.02$ | $0.41\pm0.01$ | $0.39\pm0.01$ | $0.39\pm0.01$ | $0.44\pm0.01$ | $0.44\pm0.01$ | $0.41\pm0.01$ | $0.41\pm0.01$ |
| | TGM-DLM | $0.42\pm0.01$ | $0.40\pm0.01$ | $0.42\pm0.02$ | $0.44\pm0.01$ | $0.41\pm0.01$ | $0.39\pm0.02$ | $0.46\pm0.01$ | $0.42\pm0.01$ | $0.53\pm0.01$ | $0.47\pm0.01$ |
| | HITGEN | $\mathbf{0.49\pm0.01}$ | $\mathbf{0.48\pm0.01}$ | $\mathbf{0.49\pm0.01}$ | $\mathbf{0.53\pm0.01}$ | $\mathbf{0.49\pm0.01}$ | $\mathbf{0.48\pm0.01}$ | $\mathbf{0.49\pm0.01}$ | $\mathbf{0.48\pm0.01}$ | $\mathbf{0.59\pm0.01}$ | $\mathbf{0.49\pm0.01}$ |
| RDK ↑ | GxVAEs | $0.33\pm0.01$ | $0.32\pm0.01$ | $0.33\pm0.01$ | $0.30\pm0.01$ | $0.32\pm0.01$ | $0.28\pm0.01$ | $0.35\pm0.01$ | $0.36\pm0.01$ | $0.33\pm0.01$ | $0.31\pm0.01$ |
| | TGM-DLM | $0.31\pm0.01$ | $0.31\pm0.01$ | $0.35\pm0.01$ | $0.31\pm0.01$ | $0.31\pm0.01$ | $0.30\pm0.01$ | $0.32\pm0.01$ | $0.34\pm0.01$ | $0.31\pm0.01$ | $0.30\pm0.01$ |
| | HITGEN | $\mathbf{0.37\pm0.01}$ | $\mathbf{0.36\pm0.01}$ | $\mathbf{0.37\pm0.01}$ | $\mathbf{0.37\pm0.01}$ | $\mathbf{0.35\pm0.01}$ | $\mathbf{0.34\pm0.01}$ | $\mathbf{0.35\pm0.01}$ | $\mathbf{0.36\pm0.01}$ | $\mathbf{0.43\pm0.01}$ | $\mathbf{0.35\pm0.01}$ |
| Morgan ↑ | GxVAEs | $0.12\pm0.01$ | $0.13\pm0.01$ | $0.13\pm0.01$ | $0.14\pm0.01$ | $0.13\pm0.01$ | $0.11\pm0.01$ | $0.15\pm0.01$ | $0.13\pm0.01$ | $0.14\pm0.01$ | $0.15\pm0.01$ |
| | TGM-DLM | $0.14\pm0.01$ | $0.14\pm0.01$ | $0.16\pm0.01$ | $0.13\pm0.01$ | $0.16\pm0.01$ | $0.13\pm0.01$ | $0.14\pm0.01$ | $0.13\pm0.01$ | $0.16\pm0.01$ | $0.15\pm0.01$ |
| | HITGEN | $\mathbf{0.25\pm0.01}$ | $\mathbf{0.24\pm0.01}$ | $\mathbf{0.26\pm0.01}$ | $\mathbf{0.31\pm0.01}$ | $\mathbf{0.25\pm0.01}$ | $\mathbf{0.25\pm0.01}$ | $\mathbf{0.23\pm0.01}$ | $\mathbf{0.22\pm0.01}$ | $\mathbf{0.34\pm0.01}$ | $\mathbf{0.25\pm0.01}$ |

advantage in Leven similarity further demonstrates that the designed ligands are more syntactically consistent with their reference counterparts at the character level.

HITGEN also achieves superior results in FCD and functional group Overlap, indicating that the designed targeted molecules better match the distribution of real-world ligands in terms of chemical properties. For example, under the AURKB target setting, HITGEN improves FCD and Overlap metrics by 56% and 22%, respectively, compared to the SOTA baseline, suggesting enhanced chemical validity and structural diversity tailored to AURKB.

Encouraging gains are also observed in the Tanimoto similarity metrics, particularly for the fine-grained Morgan fingerprint similarity. Across the ten targets, HITGEN delivers an average improvement of 76% over current SOTA baselines, underscoring the superior structural fidelity of its candidate ligands. Further results in Table I.4 and Figure I.5 showcase that HITGEN also dominates in terms of maximum Tanimoto similarity scores within the designed candidate molecule pool, reaffirming its advantage in producing biologically meaningful and structurally faithful molecules.

## 4.4 ABLATION STUDIES

To verify the effectiveness of HITGEN in robustly balancing transcriptome-aware biochemical information, we conduct an ablation study using several model variants. As shown in Table 3, the full HITGEN configuration achieves the best overall performance. When the fusion module is entirely removed (W/o T), performance drops across all metrics except Leven and FCD; for instance, the Total score decreases from 0.93 to 0.89. Furthermore, disabling either the chemical attention (W/o TC) or the transcriptomic attention (W/o TT) results in additional degradation across all metrics except the character-level Total score. A pronounced drop of up to 29% is observed in RDK similarity, which captures path-level molecular structural fingerprints, highlighting the crucial role of bidirectional attention in maintaining structural coherence. These findings underscore the importance of HITGEN in coordinating transcriptome profiles and expert semantic guidance to steer targeted molecule design.

To further validate the value of incorporating molecule design descriptions and optimization mechanism through human-in-the-loop manner, we conducted additional ablation studies under two scenarios: using only the transcriptome profiles to guide molecule

Table 3: Internal ablation study results of HITGEN.

| Method | Total ↑ | Leven ↓ | FCD ↓ | MACCS ↑ | RDK ↑ | Morgan ↑ |
|---|---|---|---|---|---|---|
| W/o T | 0.89 | 0.48 | 0.30 | 0.47 | 0.31 | 0.21 |
| W/o TC | 0.91 | 0.49 | 0.32 | 0.45 | 0.29 | 0.20 |
| W/o TT | 0.92 | 0.50 | 0.31 | 0.46 | 0.28 | 0.20 |
| HITGEN | **0.93** | **0.48** | **0.30** | **0.49** | **0.36** | **0.23** |

design, and removing the optimization mechanism. As shown in Appendix Table I.3, relying solely on transcriptomic information leads to a substantial decline across all metrics. For example, the fine-grained Morgan similarity decreases by 0.29; the RDK (atom-path) and MACCS (coarse-grained substructure) similarities decrease by 0.24 and 0.25, respectively; and the composite score Total also falls by more than 36%, jointly indicating a concurrent loss of structural fidelity and chem-

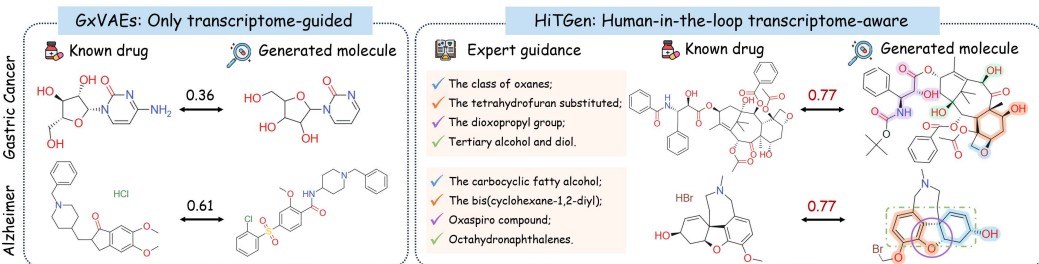

Figure 5: Comparison of therapeutic molecules generated for two specific diseases by the current SOTA GxVAEs and expert-guided HiTGen.

ical alignment. These results indicate that expert guidance offers critical chemical guidance for transcriptome-aware targeted molecule design. When the optimization mechanism is introduced, the quality of candidate molecules designed by HiTGen improves further, with an average Tanimoto similarity increase of 46%. Additionally, we verified the novelty of the optimized molecules by comparing them against the large-scale molecular corpus used to pretrain DrugAssist. The results confirm that none of the refined molecules are present in the millions of compounds used during its pretraining, ensuring structural novelty beyond the training distribution.

### 4.5 CASE STUDIES: THERAPEUTIC MOLECULE DESIGN

In disease states, transcriptomes often exhibit abnormal expression patterns, particularly in cellular environments affected by profound pathological factors such as cancer and neurodegenerative disorders. These abnormalities introduce complex perturbations across both transcriptomes and target proteins, ultimately reducing the hit rate of targeted molecules and increasing the cost of drug development. As a case study, we focus on gastric cancer and Alzheimer's disease, aiming to design chemically reasonable and structurally relevant ligands to accelerate target-specific drug discovery in these challenging therapeutic areas. To this end, we leverage patient-derived aberrant transcriptomic profiles to guide disease-aware, targeted molecule design via HiTGen.

Figure 5 illustrates the designed molecules from HiTGen compared to those produced by GxVAEs, a SOTA baseline using only transcriptome. The corresponding Morgan similarity scores to the source ligands are also presented. Under both disease conditions, where targeted therapeutics remain limited, HiTGen underscores clear structural advantages compared with GxVAEs, achieving similarity scores as high as 0.77. Notably, HiTGen adheres to the chemical intents expressed in the molecule design descriptions, validating their role in complementing disease-specific transcriptomic information. For instance, under the gastric cancer setting, all five chemical attributes outlined in the design description were faithfully reflected in the designed molecule, as marked in the highlighted substructures in Figure 5. These findings demonstrate the effectiveness of HiTGen in designing high-quality candidate molecules, serving as a promising tool for subsequent stages of structure-based drug development.

## 5 CONCLUSION

We proposed HiTGen, a novel human-in-the-loop framework for transcriptome-aware targeted molecule design. By integrating molecular textual descriptions with transcriptome profiles, HiTGen bridges biological expression and chemical design intent, enabling a more precise and controllable design process. This architecture incorporates early-stage expert guidance and molecular refinement, producing high-quality candidates aligned with both biological and chemical expectations. Extensive experiments across diverse, rigorous assessments demonstrate that HiTGen outperforms existing SOTA models. We believe HiTGen paves the way for future, large-scale collaborative research in interactive and interpretable molecule design.

Despite its promising performance, HiTGen currently relies on expert predefined design specifications, which may constrain its adaptability in real-world scenarios, particularly in complex diseases characterized by heterogeneous biological information or dynamic molecular states. In future work, we will incorporate uncertainty calibration and lightweight multi-transcriptome signals to improve robustness under distribution shifts and evolving disease states.

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

# Appendix

## A  NOTATIONS

Table A.1 compiles the mathematical notation employed in this work along with brief explanations.

Table A.1: Notation summary used in this work.

| Notations | Explanations |
|---|---|
| $E \in \mathbb{R}^K$ | Transcriptome profile (vector of $K$ expression levels induced by a drug) |
| $O \in \mathbb{R}^{M \times K}$ | Transcriptomic matrix for $M$ samples and $K$ transcriptomes |
| $F$ | Chemical specification / design intent (free-text or structured cues) |
| $R$ | Chemical semantic embedding derived from $F$. |
| $\mathcal{C} = \{c_n\}_{n=1}^N$ | VQ-VAE learnable codebook with $N$ code vectors $c_n \in \mathbb{R}^D$ |
| $X$ | Molecular representation (SELFIES token sequence). |
| $f(\cdot)$ | Decoder function (Transformer-based diffusion decoder). |
| $\mathrm{sg}(\cdot)$ | Stop-gradient operator (identity in forward pass; zero gradient backward) |
| $\mathbf{CA}(\cdot, \cdot)$ | Cross-attention mechanism mapping queries to contextual keys/values |
| $L$ | Maximum sequence length (number of tokens). |
| $\mathbf{Z}_e, \mathbf{Z}_O, \mathbf{Z}_R$ | Latent feature representations from encoder |
| $\mathbf{g}(w_j)$ | Token-embedding lookup for token $w_j$ |

## B  DATASET DETAILS

To accommodate diverse molecular generation tasks, we curate three distinct transcriptomic expression datasets, each corresponding to a unique biological condition. Prior to experimentation, all datasets undergo preprocessing to extract the top 978 landmark genes defined by the LINCS L1000 platform, followed by standardization. The datasets utilized in this study are publicly accessible and governed by licenses that permit open or research-oriented use, with detailed license terms available via their respective source repositories.

### B.1  TRANSCRIPTOMIC INDUCTIONS

**Transcriptomic Inductions** originate from the LINCS L1000 initiative and captures gene expression signatures of human cell lines exposed to a wide array of small-molecule perturbations. Each entry corresponds to a specific combination of compound, cell line, dosage, and treatment duration. For our study, we isolate data pertaining to the MCF7 cell line, encompassing expression profiles linked to 13,755 distinct compounds. These perturbation-induced signatures form the basis for modeling how transcriptomes respond to chemical stimuli, thereby supporting drug-to-expression alignment and facilitating conditional molecule generation.

### B.2  TRANSCRIPTOMIC PERTURBATIONS

**Transcriptomic Perturbations** were likewise obtained from the LINCS L1000 collection, capturing gene expression responses induced by either knockdown or overexpression of specific protein targets. In this study, we focus on ten biologically relevant targets: RAC-$\alpha$ serine/threonine-protein kinase (AKT1), RAC-$\beta$ serine/threonine-protein kinase (AKT2), Aurora B kinase (AURKB), cysteine synthase A (CTSK), epidermal growth factor receptor (EGFR), histone deacetylase 1 (HDAC1), mammalian target of rapamycin (MTOR), phosphatidylinositol 3-kinase catalytic subunit alpha (PIK3CA), decapentaplegic homolog 3 (SMAD3), and tumor protein p53 (TP53). These perturbation-induced profiles provide valuable insights into how the modulation of individual targets influences the cellular transcriptomic landscape. We leverage these data to condition molecule generation on desired target-specific effects, thereby enabling structure design guided by molecular-level intervention.

### B.3  MORBID TRANSCRIPTOMES

**Morbid Transcriptomes** were sourced from the Crowd Extracted Expression of Differential Signatures (CREEDS) database, with each disease represented by gene expression signatures encompassing 14,804 genes. In this study, we include profiles related to conditions such as gastric cancer and

---

### Expert-in-the-loop prompt

It is known that the SMILES form of the drug molecule is {original_smiles}. I would like to further optimize the structure of the following molecule to enhance its pharmacological properties and increase its Tanimoto similarity with known drug molecules.

Optional Requirements:

1. Please optimize the structure of the molecule {target_smiles}.

2. Please optimize the structure of the molecule {target_smiles} and generate two more molecules that meet my requirements.

---

Figure C.1: Prompt design used in the second-stage expert-in-the-loop optimization of HITGEN. The prompt enables structural refinement of a target molecule with optional generations.

Alzheimer's disease. For each condition, we aggregated gene expression data from multiple patient samples by averaging their profiles, resulting in a representative disease-level signature.

## C  PROMPT TEMPLATE

To incorporate human expertise into HITGEN's second-stage optimization, we adopt a prompt-driven interaction (Figure C.1) that translates design intent into constrained edits on a target molecule. The prompt anchors the search to a drug-like reference {original_smiles} and a target {target_smiles} to be refined, while allowing experts to specify optimization goals (e.g., improving task-specific pharmacology, increasing Tanimoto to the reference) and lightweight constraints (e.g., preserve core scaffold and labeled stereocenters; avoid structural alerts; maintain soft bounds on HBD/HBA, TPSA, rotatable bonds, and logP/logD). The agent then proposes one optimized revision of the target and two diverse alternates that satisfy the same intent. Each candidate is validated (RDKit sanitization/neutralization and uniqueness) and scored with fast filters (QED, SA score, TPSA, logP/logD, Tanimoto to the reference), after which the expert can tighten constraints, lock substructures, or reweight objectives for the next round.

The protocol is iterative and auditable. We maintain a compact state that records the latest accepted candidate, the set of locked substructures (SMARTS), and a short rationale for accepted edits (e.g., "raise polarity near hinge binder; retain quinazolinone core"). Diversity is encouraged by requiring the two alternates to differ from each other and from the main revision by minimal but distinct local edits (e.g., R-group swaps, isosteric ring substitutions, heteroatom repositioning), which helps explore adjacent chemotypes without incurring scaffold drift.

In practice, this prompt-driven stage acts as a fine-grained steering layer over HITGEN's first-stage outputs: it constrains modifications to chemically plausible neighborhoods, reduces unproductive exploration, and shortens the path to reference-congruent yet property-improved candidates. The design is reproducible—prompts, constraints, and scores are logged per round—and flexible: objectives can be swapped (e.g., emphasize permeability or synthetic accessibility), constraints can be relaxed to permit scaffold hopping when warranted, and the number of returned alternates can be adjusted to balance exploration and cost. Collectively, the protocol delivers short, controllable feedback loops that capture expert intent while preserving chemical realism, enabling rapid convergence toward target-aligned molecules.

## D  BASELINE MODEL DESCRIPTIONS

To rigorously assess HITGEN, we construct baselines from two complementary angles: (i) transcriptome-based deep learning models—covering conditional generation paradigms (e.g., VAE,

diffusion, autoregressive) that rely solely on gene-expression profiles—to evaluate performance under biological context only; and (ii) chemistry-intent–driven models, which condition on molecular text (design priorities, property targets, or natural-language instructions), to evaluate semantic control-lability and chemical alignment. Baselines are selected for representativeness and impact, public availability/reproducibility, strict compatibility with our conditioning setup (transcriptome or text), and coverage of diverse architectures and training strategies.

## D.1 TRANSCRIPTOME-BASED MODELS

**TRIOMRHE** is a multi-head Transformer framework designed to model the relationship between transcriptome data and molecular responses. It leverages heterogeneous embedding strategies and attention mechanisms to map gene expression to molecular representations. We follow the official configuration and evaluate on our benchmark settings.

**GxRNN** is a transcriptome-guided sequence generation model based on recurrent neural networks. It takes preprocessed gene expression vectors as input and autoregressively generates SMILES sequences. We re-implement the model using the settings described in the original paper.

**HNN2Mol** is a hybrid neural network that integrates gene expression and pathway-level features for molecule generation. It combines feedforward and convolutional layers for representation learning. We apply this model using their released code and apply it to our normalized data.

**GxVAEs** is a variational autoencoder model conditioned on gene expression, trained to reconstruct molecules from transcriptomic input. It uses a shared latent space to couple gene expression vectors and molecular structures. We use their pretrained weights and apply our gene profiles to generate molecules.

## D.2 CHEMISTRY-INTENT–DRIVEN MODELS

**Mol-T5** is a Transformer-based large language model pretrained on molecular SMILES. We use the mol-t5-base checkpoint and directly prompt the model with target-specific textual descriptions. No additional fine-tuning is performed.

**Text+ChemT5** combines natural language input with chemical prompts using the ChemT5, which has been pretrained on both molecule text and structural representations. We concatenate molecular textual descriptions with target information and use the model in zero-shot mode.

**TGM-DLM** is a two-stage diffusion-based framework for text-guided molecule generation. The first stage generates a coarse molecular scaffold from a textual description, and the second stage refines it into a complete and valid molecule through conditional diffusion decoding. Both stages leverage denoising diffusion models guided by language-derived embeddings. We adopt the official code for our evaluations.

**Mol-Instructions** is an instruction-tuned molecular language model fine-tuned on various molecule-related generation tasks using instruction–response formats. We adopt the official prompt format and apply it to conditional molecule generation tasks without additional supervision.

## E  METRIC DETAILS

To offer a well-rounded evaluation of the HITGEN framework, this section outlines the computational procedures for all employed metrics, including statistical indicators, molecular structural properties, and functional overlap rate. Together, these metrics capture complementary aspects of the generated molecules—covering their quality, diversity, and biological relevance—and provide an integrated assessment of the model's overall performance.

**Validity.**  This metric quantifies the proportion of chemically valid molecules within the generated set $\mathcal{G}$. A molecule $g_i \in \mathcal{G}$ is deemed valid if it passes chemical integrity checks, such as sanitization via RDKit or successful decoding from SELFIES. Let $\mathcal{V} \subseteq \mathcal{G}$ denote the set of valid molecules. The validity score is given by:

$$\text{Validity} = \frac{|\mathcal{V}|}{|\mathcal{G}|}$$

**Uniqueness.**  Uniqueness measures the diversity of valid molecules by computing the fraction of non-duplicate entries in $\mathcal{V}$. Denoting the set of distinct valid molecules as $\text{unique}(\mathcal{V})$, the uniqueness score is defined as:

$$\text{Uniqueness} = \frac{|\text{unique}(\mathcal{V})|}{|\mathcal{V}|}$$

**Novelty.**  Novelty evaluates the ability of the model to generate molecules outside the training distribution. It is calculated as the proportion of generated samples not found in the training set $\mathcal{D}_{\text{train}}$:

$$\text{Novelty} = \frac{|\{g \in \mathcal{G} \mid g \notin \mathcal{D}_{\text{train}}\}|}{|\mathcal{G}|}$$

**Levenshtein Similarity.**  To evaluate the string-level similarity between generated molecules and their corresponding references, we compute the normalized Levenshtein distance over a fixed scale. For each molecule–reference pair $(g_i, r_i)$, let $d(g_i, r_i)$ be the raw edit distance between their string representations. This distance is linearly normalized by a predefined upper bound (50 in our case), and the similarity is defined as:

$$\text{LevSim}(g_i, r_i) = 1 - \frac{d(g_i, r_i)}{50}$$

The final Levenshtein similarity score is the mean of all pairwise similarities:

$$\text{Levenshtein Similarity} = \frac{1}{N} \sum_{i=1}^{N} \text{LevSim}(g_i, r_i)$$

**Fréchet ChemNet Distance (FCD).**  FCD assesses how closely the generated distribution $\mathcal{G}$ aligns with the reference set $\mathcal{R}$ in a learned chemical embedding space. Specifically, it computes the Fréchet distance between two multivariate Gaussians fitted to the activations of a pretrained ChemNet model. Let $(\mu_g, \Sigma_g)$ and $(\mu_r, \Sigma_r)$ denote the mean and covariance of molecular embeddings, then:

$$\text{FCD} = \|\mu_g - \mu_r\|^2 + \text{Tr}\left(\Sigma_g + \Sigma_r - 2\left(\Sigma_g \Sigma_r\right)^{1/2}\right)$$

**Morgan Tanimoto Similarity.**  This metric calculates the average Tanimoto similarity between generated and reference molecules based on ECFP4 fingerprints, which capture topological features around each atom. Given binary fingerprints $\mathbf{f}_g$ and $\mathbf{f}_r$, the similarity is computed as:

$$\text{Tanimoto}(\mathbf{f}_g, \mathbf{f}_r) = \frac{|\mathbf{f}_g \cap \mathbf{f}_r|}{|\mathbf{f}_g \cup \mathbf{f}_r|}$$

**RDK Tanimoto Similarity.**  This metric computes the Tanimoto similarity between generated and reference molecules using RDKit topological fingerprints, which encode molecular paths based on atom connectivity. Given binary fingerprints $\mathbf{r}_g$ and $\mathbf{r}_r$ for a generated molecule and its corresponding reference, respectively, the similarity is calculated as:

$$\text{Tanimoto}(\mathbf{r}_g, \mathbf{r}_r) = \frac{|\mathbf{r}_g \cap \mathbf{r}_r|}{|\mathbf{r}_g \cup \mathbf{r}_r|}$$

Compared to other fingerprints, RDK-based fingerprints emphasize linear connectivity and local substructure arrangements, offering a complementary perspective on structural similarity.

**MACCS Tanimoto Similarity.**  Analogous to the Morgan metric, this variant uses MACCS keys, a predefined substructure-based fingerprint. For binary vectors $\mathbf{m}_g$ and $\mathbf{m}_r$, the Tanimoto similarity is:

$$\text{Tanimoto}(\mathbf{m}_g, \mathbf{m}_r) = \frac{|\mathbf{m}_g \cap \mathbf{m}_r|}{|\mathbf{m}_g \cup \mathbf{m}_r|}$$

This provides a coarser-grained estimate of structural similarity based on substructure patterns.

**Functional Group Overlap Rate.** To evaluate the chemical functional consistency between generated molecules and their reference counterparts, we compute a Functional Group (FG) Overlap Rate based on RDKit-defined SMARTS patterns.

First, we apply RDKit's built-in substructure matching to extract functional groups from all molecules in the training set. Each molecule is scanned against a curated library of SMARTS patterns covering common chemical functionalities (e.g., amines, carboxylic acids, sulfonamides). We then rank all identified functional groups by their frequency and select the top 32 most prevalent ones as the evaluation set.

For a given molecule $x$, we construct a binary vector $\mathbf{f}(x) \in \{0, 1\}^{32}$, where each dimension indicates the presence or absence of a specific functional group from the top-32 list. Given a generated molecule $g_i$ and its corresponding reference molecule $r_i$, their functional group overlap is computed as the Jaccard index:

$$\text{FGOverlap}(g_i, r_i) = \frac{|\mathbf{f}(g_i) \cap \mathbf{f}(r_i)|}{|\mathbf{f}(g_i) \cup \mathbf{f}(r_i)|}$$

The final FG Overlap Rate is averaged over all molecule pairs in the evaluation set:

$$\text{Functional Group Overlap Rate} = \frac{1}{N} \sum_{i=1}^{N} \text{FGOverlap}(g_i, r_i)$$

This metric captures the extent to which the generated molecules retain or reproduce key functional groups present in the reference molecules, reflecting alignment at the substructure level.

## F  IMPLEMENTATION DETAILS

This section provides a comprehensive account of our experimental implementation, including model architecture and training configurations, hyperparameter choices and reproducibility settings (random seeds and hardware/software environment). We also document the paper-writing and visualization workflow—including how LLMs were used and subsequently human-verified—to ensure transparency and reproducibility.

### F.1  EXPERIMENTAL FACILITIES

HITGEN is developed using the PyTorch framework and executed on an NVIDIA GeForce RTX 3090 GPU with an Intel i7-9700K CPU. All components are implemented in Python 3.8. In module (c) *Profile Refiner*, both encoder and decoder adopt three-layer feedforward networks with hidden dimensions of 512, 256, and 128, respectively. The model is optimized with a learning rate of $1e-4$ and a dropout rate of 0.2. In module (d) *Expert Posterior*, the SELFIES representations are tokenized with a maximum sequence length of 256, and the embedding dimension of the learnable tokens is set to 32. A frozen textual encoder with a fixed embedding size of 768 is used to extract features from molecular descriptions. For module (f) *Molecule Optimization*, the diffusion model is configured with $T = 2000$ denoising steps, a learning rate of $1e-4$, and a dropout of 0.1. To improve sampling efficiency during inference, we apply a uniform skip-sampling strategy to the remapped SELFIES sequences. Finally, in module (g) *Fallback Strategy Filtering*, any chemically invalid outputs are removed, and the remaining valid candidate molecules from the diffusion process are retained for downstream evaluation.

### F.2  LLM USAGE STATISTICS

We used large language models solely for language polishing (grammar, wording, and style) of author-written text. All suggestions were manually reviewed and edited by the authors. LLMs did not contribute to technical content or decision making (including method design, experimental setup, result analysis, conclusions, or code), nor were they used to generate synthetic data, figures, or citations. To ensure privacy and compliance, we did not provide the models with any unpublished data, personally identifiable information, or restricted materials. The authors take full responsibility for the content and any remaining errors.

Table G.2: Performance comparison of expert-in-the-loop molecule optimization guided by Drug Assistant. HITGEN widely outperforms baseline models.

| Models | Total↑ | Validity(%)↑ | Unique(%)↑ | Novelty(%)↑ | Levenshtein↓ | FCD↓ | MACCS↑ | RDK↑ | Morgan↑ |
|---|---|---|---|---|---|---|---|---|---|
| Mol-T5 | 0.65 | 78.92 | 81.86 | 99.89 | 1.04 | 0.70 | 0.34 | 0.24 | 0.10 |
| Text+ChemT5 | 0.62 | 63.59 | 98.99 | 98.13 | 0.66 | 0.34 | 0.44 | 0.34 | 0.27 |
| Mol-Instructions | 0.83 | 86.93 | 95.57 | **100.0** | 0.78 | 0.39 | 0.28 | 0.18 | 0.10 |
| HITGEN used | **0.98** | 100.0 | **99.71** | 98.47 | **0.45** | **0.17** | **0.63** | **0.51** | **0.40** |

## G  HITGEN WITH HUMAN-IN-THE-LOOP ADVANTAGE

### G.1  COMPARISON OF DOMAIN OPTIMIZATION LLMS

To evaluate the effectiveness of human-in-the-loop refinement, we compare HITGEN with existing text-guided molecule optimization baselines, including Mol-T5, Mol-Instructions, and ChemT5-augmented prompts. As shown in Table G.2, HITGEN achieves consistently superior performance across all evaluation metrics. In particular, it yields the highest overall score (0.98), perfect validity (100%), and the best chemical similarity scores (e.g., FCD = 0.17, Morgan = 0.40).

Notably, while Mol-Instructions and Mol-T5 exhibit high novelty, they suffer from substantial degradation in structure-level metrics. For instance, Mol-T5 shows the worst Levenshtein similarity (1.04) and a relatively low MACCS similarity (0.34), indicating that its generated molecules deviate significantly from target references despite being novel. Similarly, Mol-Instructions, although producing valid outputs, underperforms in RDK and Morgan similarity, implying reduced structural and substructure alignment.

These results demonstrate that naïvely applying general-purpose LLMs (e.g., Mol-T5) or instruction-tuned chemical models does not guarantee effective molecular optimization. In contrast, our HITGEN framework, augmented by DrugAssist feedback in the second stage, enables controlled structure refinement while preserving chemical relevance and diversity. This highlights the importance of expert-in-the-loop mechanisms in achieving successful optimization in chemically grounded generation tasks.

### G.2  TEXTUAL OVERLAP

To better understand the structural characteristics of molecules in the test set, we conducted a frequency analysis of functional groups and visualized the top 32 most frequent ones (see Figure G.2). As shown, the most prevalent groups include aromatic (1303 occurrences), phenyl (1236), and benzene (1236), highlighting the dominance of aromatic ring structures in the dataset, which is consistent with their widespread presence in drug-like compounds.

Other commonly observed functional groups include amide (556), carbonyl (803), amine (413), and hydroxyl (383), indicating the importance of polar functionalities in this context. In contrast, groups such as guanidine (12) and thiol (1) appear rarely, suggesting their limited representation and potential difficulty for the model to learn or generalize over these patterns.

These statistics ground the interpretation of downstream evaluations—e.g., functional-group reconstruction and diversity analysis—particularly for rare or underrepresented motifs. To assess the chemical relevance and text–structure alignment of generated molecules, we measure functional-group overlap in a text-based space. Concretely, for each generated and reference molecule we: (i) render a concise natural-language description; (ii) map tokens/phrases to a controlled functional-group lexicon linked to RDKit-defined SMARTS; and (iii) instantiate a set of functional groups per molecule. Overlap metrics (e.g., Jaccard, precision/recall) are then computed between these sets, yielding a semantic substructure–level comparison that is robust to synonymy in descriptions while remaining chemically grounded by the underlying SMARTS definitions.

Method (pseudocode overview). Our counting pipeline follows **Algorithm G.1**: (i) *Preprocess*—compile RDKit SMARTS patterns for a curated set of functional groups; (ii) *Extractor*—for a given SMILES, return the set of groups detected via substructure matches; (iii) *Iterate*—scan all test-set molecules, accumulate per-group totals; (iv) *Rates/Sorting*—rank groups by frequency and (when needed) compute per-group match rates; (v) *Aggregates*—report weighted averages over

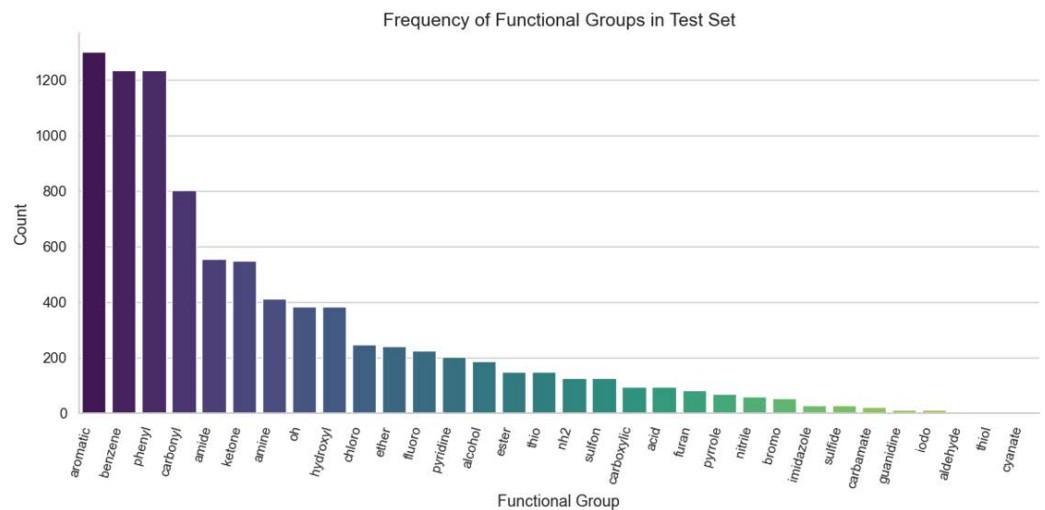

Figure G.2: Frequency distribution of functional groups in the test set.

---

**Algorithm G.1** Functional-Group Matching with SMARTS (Structured)

---

**Require:** FG map $\mathcal{F}$ (name→SMARTS); CSV columns ($s_1 = \text{ref}, s_2, s_4$)
**Ensure:** Counts $C_{\text{tot}}$, rates $\text{rate2}, \text{rate4}$; Top-$K$/All averages
1: $\mathcal{P} \leftarrow \{(f, \text{MolFromSmarts}(\mathcal{F}[f]))\}$; $\quad C_{\text{tot}}, C_2, C_4 \leftarrow 0$
2: **function** FG($s$)
3: $\quad M \leftarrow \text{MolFromSmiles}(s)$
4: $\quad$ **return** $\varnothing$ **if** $M$ fails **else** $\{ f \mid (f,p) \in \mathcal{P}, \ M \text{ matches } p \}$
5: **end function**
6: **for all** row ($s_1, s_2, s_4$) **do**
7: $\quad \mathcal{S}_1 \leftarrow \text{FG}(s_1)$ **; continue if** $\mathcal{S}_1 = \varnothing$
8: $\quad \mathcal{S}_2 \leftarrow \text{FG}(s_2)$; $\quad \mathcal{S}_4 \leftarrow \text{FG}(s_4)$ **if valid else** $\mathcal{S}_2$
9: $\quad$ **for all** $f \in \mathcal{S}_1$ **do**
10: $\quad\quad C_{\text{tot}}[f] \leftarrow C_{\text{tot}}[f] + 1; C_2[f] \leftarrow C_2[f] + \mathbb{K}[f \in \mathcal{S}_2]; C_4[f] \leftarrow C_4[f] + \mathbb{K}[f \in \mathcal{S}_4]$
11: $\quad$ **end for**
12: **end for**
13: **for all** $f$ **s.t.** $C_{\text{tot}}[f] > 0$ **do**
14: $\quad \text{rate2}[f] \leftarrow C_2[f]/C_{\text{tot}}[f]; \quad \text{rate4}[f] \leftarrow C_4[f]/C_{\text{tot}}[f]$
15: **end for**
16: Let $\mathcal{N}$ be FGs sorted by $C_{\text{tot}}$; $\mathcal{A}_8 \leftarrow \mathcal{N}[1\text{:}8], \mathcal{A}_{16} \leftarrow \mathcal{N}[1\text{:}16], \mathcal{A}_{\text{all}} \leftarrow \mathcal{N}$
17: **for** $k \in \{2, 4\}$ **do**
18: $\quad$ **for all** $\mathcal{A} \in \{\mathcal{A}_8, \mathcal{A}_{16}, \mathcal{A}_{\text{all}}\}$ **do**
19: $\quad\quad \overline{r}_k(\mathcal{A}) \leftarrow \frac{\sum_{f \in \mathcal{A}} C_k[f]}{\sum_{f \in \mathcal{A}} C_{\text{tot}}[f]}$
20: $\quad$ **end for**
21: **end for**

---

Top-8/Top-16/All groups. This structured procedure underpins the statistics in Figure G.2 and ensures reproducible frequency estimates across SMARTS-defined motifs.

Figure G.3 presents the average FG overlap between generated and reference molecule descriptions across four evaluation sets: a specific ether-based category, and the Top-8, Top-16, and Top-32 most frequently occurring functional groups. Our proposed method, HITGEN, achieves the highest overlap across all settings—surpassing both the baseline TGM-DLM and its single-stage ablation variant HITGEN (w/o O). In particular, HITGEN attains an overlap of 0.41 and 0.38 in the Top-8 and Top-16 settings, respectively, demonstrating its ability to retain chemically meaningful substructures in both molecular structure and corresponding textual outputs.

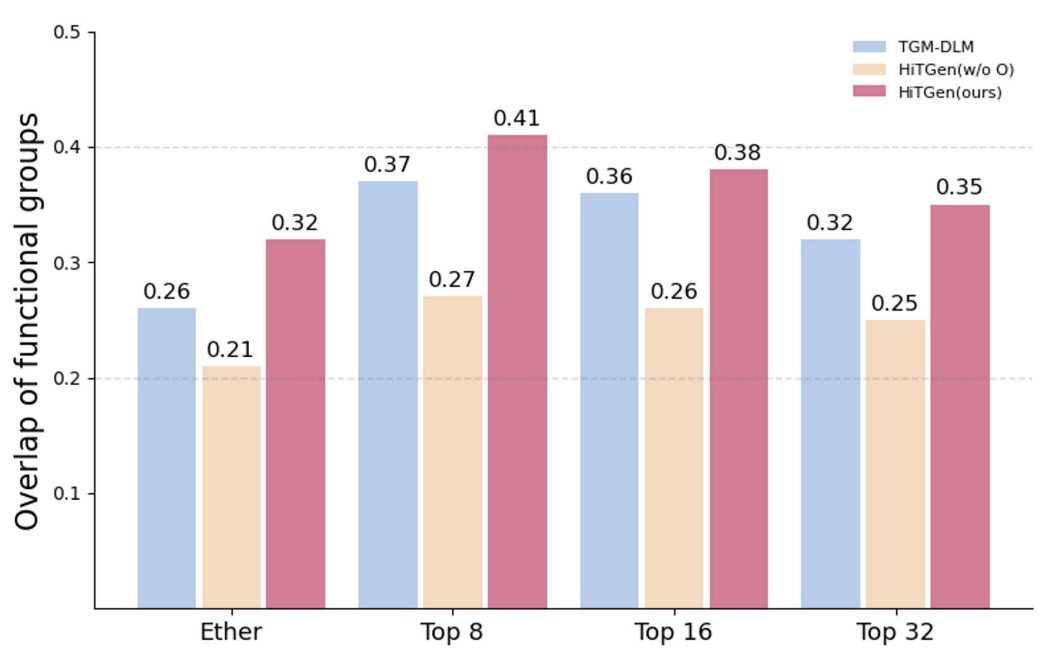

Figure G.3: Text functional group overlap between generated molecules and reference molecules across different evaluation sets: ether subgroup, top-8, top-16, and top-32 most frequent functional groups.

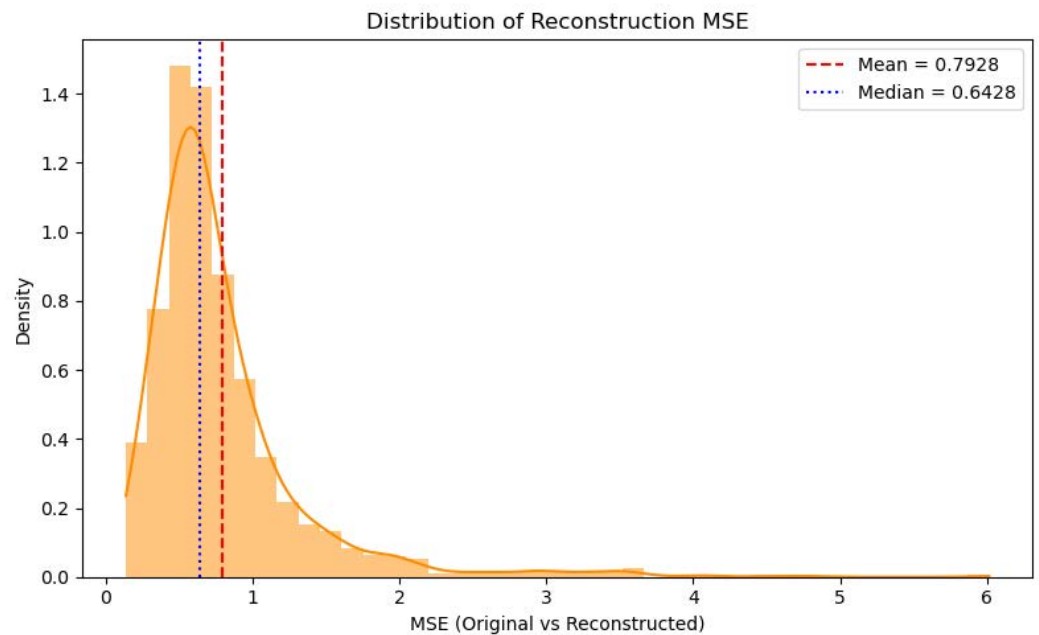

Figure H.4: Distribution of reconstruction MSE between original and decoded transcriptomes using VQ-VAE. The red dashed line indicates the mean, and the blue dotted line indicates the median.

Table I.3: Ablation study results of HITGEN under different component removals.

| Models | Total↑ | Validity(%)↑ | Unique(%)↑ | Novelty(%)↑ | Levenshtein↓ | FCD↓ | MACCS↑ | RDK↑ | Morgan↑ |
|---|---|---|---|---|---|---|---|---|---|
| HITGEN$_{w/o\ T}$ | 0.72 | 100.0 | 85.25 | 84.75 | 0.66 | 0.84 | 0.38 | 0.27 | 0.11 |
| HITGEN$_{w/o\ A}$ | 0.90 | 100.0 | 99.06 | 90.49 | 0.48 | 0.30 | 0.47 | 0.31 | 0.21 |
| HITGEN$_{w/o\ O}$ | 0.93 | 100.0 | 99.20 | 93.93 | 0.48 | 0.30 | 0.49 | 0.36 | 0.24 |
| HITGEN | **0.98** | 100.0 | **99.71** | **98.47** | **0.45** | **0.17** | **0.63** | **0.51** | **0.40** |

## H  ABLATION STUDIES

### G.1  TRANSCRIPTOMIC RECONSTRUCTION VIA VQ-VAE

To evaluate the representational capacity of our VQ-VAE encoder–decoder architecture on transcriptomic data, we examine the reconstruction error between the original and decoded gene expression profiles. Specifically, we calculate the mean squared error (MSE) across all samples in the test set and visualize the distribution in Figure H.4.

The histogram reveals that the majority of reconstruction MSE values are concentrated below 1.5, with a pronounced peak around 0.6. The mean and median MSE are 0.7928 and 0.6428, respectively, indicating a slight positive skew. This suggests that while most reconstructed profiles closely resemble their originals, a small number of outliers exhibit higher deviation.

Overall, these results demonstrate that the VQ-VAE model effectively captures the underlying patterns of the transcriptome, enabling the learned latent representations to preserve fine-grained biological signals essential for downstream generation and optimization tasks.

### G.2  ADDITIONAL ABLATION STUDY

To evaluate the contribution of individual components within the HITGEN framework, we perform external ablation experiments under three variant settings:

- **HITGEN$_{w/o\ T}$**: removes the expert-provided textual molecule description, eliminating high-level guidance from the prompt. *Without textual intent, the model must rely solely on transcriptomic signals and generic priors, typically reducing substituent specificity and scaffold selection fidelity.*

- **HITGEN$_{w/o\ A}$**: disables the bidirectional cross-attention module, thus decoupling the integration between transcriptomic and textual modalities. *In the absence of explicit cross-modal fusion, conditioning collapses to weaker late or independent cues, leading to poorer alignment between modalities and degraded control over local topology.*

- **HITGEN$_{w/o\ O}$**: excludes the second-stage human-in-the-loop optimization, relying solely on one-shot molecule generation. *This ablation removes iterative expert steering, often increasing scaffold drift and lowering target similarity/property satisfaction compared with the full two-stage pipeline.*

Table I.3 reports the quantitative performance across several molecular generation metrics. The full HITGEN model achieves the best overall results, including the highest Total Score (0.98), Validity (100%), Uniqueness (99.71%), and Novelty (98.47%), while also minimizing distance-based metrics such as Levenshtein (0.45) and FCD (0.17).

Compared to the full model, removing the textual prompt (*w/o T*) significantly reduces generation diversity and novelty, indicating the importance of expert-level semantic guidance in improving candidate quality. Disabling the attention fusion mechanism (*w/o A*) leads to moderate degradation in structure similarity scores (e.g., MACCS: 0.49 → 0.63 in full model), suggesting the cross-modal integration plays a key role in aligning molecular structure with biological context. Finally, removing the second-stage interactive optimization (*w/o O*) yields the largest drop in performance across almost all dimensions, especially in overlap-sensitive scores such as Morgan and RDK similarity. This highlights the critical role of human-in-the-loop refinement in enhancing pharmacological relevance and specificity.

Table I.4: Target-specific comparison of maximum Tanimoto similarity using different fingerprinting methods (MACCS, RDK, Morgan). HITGEN achieves the highest structural alignment with reference molecules across all targets.

| Metrics | Methods | AKT1 | AKT2 | AURKB | CTSK | EGFR | HDAC1 | MTOR | PIK3CA | SMAD3 | TP53 |
|---|---|---|---|---|---|---|---|---|---|---|---|
| MACCS ↑ | GxVAEs | 1.00 | 0.78 | 0.78 | 0.75 | 0.87 | 0.80 | 0.83 | 0.78 | 0.94 | 0.94 |
| | TGM-DLM | 0.81 | 0.75 | 0.79 | 0.72 | 0.80 | 0.74 | 0.82 | 0.79 | 0.96 | 0.86 |
| | HITGEN | **1.00** | **0.98** | **0.98** | **0.99** | **0.98** | **0.98** | **0.99** | **0.98** | **1.00** | **1.00** |
| RDK ↑ | GxVAEs | 1.00 | 0.66 | 0.70 | 0.60 | 0.80 | 0.66 | 0.61 | 0.55 | 0.92 | 0.86 |
| | TGM-DLM | 0.52 | 0.51 | 0.58 | 0.49 | 0.75 | 0.64 | 0.57 | 0.52 | 0.81 | 0.83 |
| | HITGEN | **1.00** | **0.99** | **0.98** | **0.99** | **0.98** | **0.97** | **0.98** | **0.99** | **1.00** | **1.00** |
| Morgan ↑ | GxVAEs | 0.85 | 0.43 | 0.47 | 0.38 | 0.74 | 0.55 | 0.52 | 0.35 | 0.98 | 0.76 |
| | TGM-DLM | 0.47 | 0.39 | 0.46 | 0.39 | 0.50 | 0.46 | 0.42 | 0.42 | 0.83 | 0.73 |
| | HITGEN | **0.91** | **0.93** | **0.82** | **0.99** | **0.95** | **0.92** | **0.87** | **0.90** | **1.00** | **0.97** |

Figure I.5: Detailed illustration of HITGEN's performance in terms of biological morgan Tanimoto similarity under ten ligands.

## I EVALUATION RESULTS OF HITGEN-DESIGNED MOLECULES

To further assess target-specific structural fidelity, we report the maximum Tanimoto similarity for each generated molecule under three complementary fingerprints—MACCS (substructure keys), RDK (path-based), and Morgan (circular/topological) (Table I.4). Higher scores reflect stronger alignment from coarse fragment overlap (MACCS) to path consistency (RDK) and fine-grained local topology (Morgan).

Overall performance. HITGEN attains the best or tied-best score on every target across all three fingerprints. Under MACCS, HITGEN is near-saturated (0.98–1.00) and reaches 1.00 on multiple targets (AKT1, SMAD3, TP53), with large margins over GxVAEs/TGM-DLM (e.g., AKT2: 0.98 vs. 0.78/0.75; MTOR: 0.99 vs. 0.83/0.82), except for a tie on AKT1 where GxVAEs also attains 1.00. The advantage persists and widens under RDK, where HITGEN scores 0.97–1.00, again tying GxVAEs on AKT1 (1.00) but substantially surpassing both baselines elsewhere (e.g., PIK3CA: 0.99 vs. 0.55/0.52; MTOR: 0.98 vs. 0.61/0.57). On the more discriminative Morgan fingerprints, HITGEN

maintains high alignment (0.82–1.00), notably CTSK: 0.99 vs. 0.38/0.39, AKT2: 0.93 vs. 0.43/0.39, MTOR: 0.87 vs. 0.52/0.42, and SMAD3: 1.00 vs. 0.98/0.83.

**What the three fingerprints tell us.** The MACCS results indicate that HITGEN reliably recovers key fragments and functional groups. RDK further shows improved path-level substitution patterns—HITGEN matches the reference connectivity more faithfully than baselines for most targets. Morgan highlights the most stringent aspect—local neighborhoods/topology and stereochemical context—where HITGEN's margins on difficult targets (e.g., AURKB, MTOR, PIK3CA) suggest better preservation of substituent placement and local ring environments rather than merely sharing coarse fragments.

Figure I.5 provides qualitative evidence that HITGEN better preserves reference-congruent scaffolds and complex substitution patterns compared with the baselines. The side-by-side visualizations echo the quantitative trends: HITGEN preserves reference-congruent scaffolds and reconstructs complex motifs (multi-ring/heteroaromatic cores, ring fusions, and strategically positioned heteroatoms) while maintaining correct substitution patterns. In contrast, GxVAEs and TGM-DLM more frequently drift to simpler aromatics or misplace heteroatoms/linkers, which aligns with their lower RDK/Morgan maxima—especially on MTOR and AURKB, where local topology matters.

As a whole, Table I.4 and Figure I.5 show that HITGEN delivers multi-scale structural fidelity—from fragment presence (MACCS) to substitution paths (RDK) and local topology (Morgan)—yielding molecules that are not only valid and novel but also closely tailored to the intended targets. Occasional ties (e.g., AKT1 under MACCS/RDK) aside, the consistent margins—particularly on Morgan—underscore the benefit of combining transcriptomic guidance with expert-driven optimization to recover target-relevant chemotypes without collapsing structural complexity.

