# OpenReview forum: "Human-in-the-Loop Targeted Molecule Design Informed by Transcriptomes"
_ICLR.cc/2026/Conference — Submitted to ICLR 2026_

### Official Review · Reviewer_tpQJ · 2025-10-26

**Soundness:** 2
**Presentation:** 1
**Contribution:** 2
**Rating:** 2
**Confidence:** 3

**Summary:**

The manuscript proposes HITGEN, a framework for targeted molecule design. HITGEN integrates transcriptomic information and molecular descriptions, where both modalities are transformed into a latent space via encoders, followed by a decoder for molecule generation. The authors further introduce a human-in-the-loop architecture based on DrugAssist to enable fine-grained molecule optimization through designed prompting. Experimental results demonstrate that HITGEN outperforms baseline methods across multiple evaluation metrics.

**Strengths:**

- The manuscript presents an interesting attempt to combine multimodal molecular representations with a human-in-the-loop optimization strategy.
- The experimental results indicate improved performance compared to baseline methods.

**Weaknesses:**

- Writing clarity: Certain descriptions are unclear. For instance, in the sentence “Given a natural language description F that reflects the chemical specification associated with the O”, the variable F is not explained. The illustration in Figure 1 is also confusing — Step 1(b) and Step 3 both appear to involve human-in-the-loop processes, making the workflow unclear.
- Limited novelty: Although the manuscript highlights the human-in-the-loop component as a key contribution, this part primarily builds on the existing DrugAssist framework, which limits the originality of the proposed approach.
- Unclear design choices: Some methodological decisions lack sufficient justification. For example:
    - Why does the model use discrete tokens in the latent space instead of continuous representations?
    - In the semantic embedding encoder, why is the masking applied only to tokens lacking chemical relevance?

**Questions:**

- In the semantic embedding encoder, why is masking applied only to tokens that lack chemical relevance?
- How does the model perform without the DrugAssist based optimization module?

---

> ### Author Response · Authors · 2025-11-20
> **Response to the reviewer tpQJ (01)**
>
> Thank you for taking the time to review our work and for providing important suggestions. We highly value your feedback and offer the following responses and clarifications:
>
> **Response W1+W2: **
>
> **A two-stage expert-in-the-loop workflow guides both initial generation and refinement**
>
> We will further improve the clarity of the manuscript and are actively revising the relevant descriptions. Regarding the notation issue, the variable F appears only once and directly follows the phrase “natural language description,” so we did not provide additional notation; however, we agree that an explicit definition would improve readability and will clarify this in the revision.
>
> Concerning the workflow illustration, HiTGen’s human-in-the-loop interactions occur in two distinct phases. In Stage I, the molecular descriptions are automatically generated to simulate how chemists articulate structural features and functional groups, and all generated descriptions are reviewed and corrected by two medicinal chemists before being used. In Stage II, chemists propose optimization directions; an AI specialist then translates these intentions into prompts, which are subsequently reviewed by the chemists before being used to refine the generated molecules. We will revise Figure 1 to more clearly distinguish these two phases. We hope this resolves the confusion.

---

> > ### Author Response · Authors · 2025-11-20
> > **Response to the reviewer tpQJ (02)**
> >
> > **Response Q1+W3:**
> >
> > **Why discrete latent tokens?**
> >
> > We adopt a discrete latent representation because molecular structures are inherently symbolic and compositional. Discrete codes provide a stable way to capture functional groups, scaffolds, and other structural motifs, while avoiding the mode collapse and semantic drift commonly observed in continuous latent diffusion. They also enable more controllable editing and optimization, as the latent space aligns naturally with SMILES/SELFIES-like syntax.
> >
> > This design choice is consistent with recent advances in Diffusion Language Models, which demonstrate that discrete token-based diffusion offers superior controllability and semantic stability compared to continuous alternatives. For example, Diffusion-LM improves controllable text generation by leveraging discrete latent sequences rather than continuous noise vectors [1]. In molecular domains, text-guided diffusion LMs similarly operate in discrete token spaces to better capture chemical constraints and symbolic structure, as shown in text-guided molecule generation [2] and multi-property molecular optimization [3].
> >
> > These findings collectively support our decision to use a discrete latent space, and we will expand this justification in the revision.
> >
> > **Why mask only non-chemical tokens? **
> >
> > The masking strategy is designed to emphasize chemically meaningful cues while reducing noise from irrelevant linguistic tokens. Prior work in domain-specific language modeling has shown that general-purpose tokens often dilute domain semantics: LinkBERT [4] highlights that models benefit from focusing attention on semantically informative units rather than treating all tokens uniformly; BioBERT [5] similarly observes that stopwords and connective phrases contribute little biomedical meaning and can introduce noise; and MolT5 [6] emphasizes that chemically meaningful units—such as functional groups and structural descriptors—should dominate the representation space over generic natural-language fillers.
> >
> > Motivated by these findings, our masking removes only non-chemical tokens (e.g., stopwords, connectives, grammatical fillers), ensuring that attention is concentrated on chemistry-relevant terms. This preserves the full informational content of chemically meaningful expressions while preventing spurious correlations arising from general-language structure. As a result, the semantic embedding becomes more stable and provides more reliable conditioning signals for molecule generation. We will clarify this rationale in the revision.
> >
> > **References**
> >
> > [1] Li X, Thickstun J, Gulrajani I, et al. Diffusion-lm improves controllable text generation[J]. Advances in neural information processing systems, 2022, 35: 4328-4343.
> >
> > [2] Gong H, Liu Q, Wu S, et al. Text-guided molecule generation with diffusion language model[C]//Proceedings of the AAAI Conference on Artificial Intelligence. 2024, 38(1): 109-117.
> >
> > [3] Xiong Y, Li K, Chen J, et al. Text-guided multi-property molecular optimization with a diffusion language model[J]. arXiv preprint arXiv:2410.13597, 2024.
> >
> > [4] Yasunaga M, Leskovec J, Liang P. LinkBERT: Pretraining Language Models with Document Links[C]//Proceedings of the 60th Annual Meeting of the Association for Computational Linguistics (Volume 1: Long Papers). 2022: 8003-8016.
> >
> > [5] Lee J, Yoon W, Kim S, et al. BioBERT: a pre-trained biomedical language representation model for biomedical text mining[J]. Bioinformatics, 2020, 36(4): 1234-1240.
> >
> > [6] Edwards C, Lai T, Ros K, et al. Translation between Molecules and Natural Language[C]//Proceedings of the 2022 Conference on Empirical Methods in Natural Language Processing. 2022: 375-413.

---

> > > ### Author Response · Authors · 2025-11-20
> > > **Response to the reviewer tpQJ (03)**
> > >
> > > **Response Q2:**
> > >
> > > **Table I.3: Ablation study results of HiTGen under different component removals**
> > >
> > > | Model        | Total↑ | Validity(%)↑ | Unique(%)↑ | Novelty(%)↑ | Levenshtein↓ | FCD↓ | MACCS↑ | RDK↑ | Morgan↑ |
> > > |--------------|--------|--------------|------------|-------------|--------------|------|--------|------|---------|
> > > | HiTGen w/o T(text) | 0.72   | 100.0        | 85.25      | 84.75       | 0.66         | 0.84 | 0.38   | 0.27 | 0.11    |
> > > | HiTGen w/o P (transcriptome profile) | 0.71   | 100.0        | 83.06      | 85.49       | 0.69         | 0.85 | 0.35   | 0.26 | 0.11    |
> > > | HiTGen  w/o A (attention) | 0.90   | 100.0        | 99.06      | 90.49       | 0.48         | 0.30 | 0.47   | 0.31 | 0.21    |
> > > | HiTGen w/o O | 0.93   | 100.0        | 99.20      | 93.93       | 0.48         | 0.30 | 0.49   | 0.36 | 0.24    |
> > > | **HiTGen**   | **0.98** | **100.0**      | **99.71**    | **98.47**     | **0.45**       | **0.17** | **0.63**   | **0.51** | **0.40**    |
> > >
> > > The results without the DrugAssist-based optimization module are reported in Appendix Table I.3 (p.22) as HiTGen w/o O, reflecting performance when no optimization is applied. For completeness, Table G.2 reports results where DrugAssist is replaced by other domain LLMs (Mol-T5, Text+ChemT5, Mol-Instructions); all alternatives perform substantially worse. Together, these tables show both sides of the ablation: (1) HiTGen without optimization, and (2) optimization replaced by other LLMs.
> > >
> > > DrugAssist does not interpret transcriptome expression profiles—it only refines the candidates produced by HiTGen’s Stage I.
> > >
> > > 1.	**DrugAssist cannot generate molecules from gene-expression data**
> > >
> > > Existing LLM-based optimizers do not interpret transcriptomic signatures and therefore cannot perform the task gene expression → molecule generation. This capability is provided only by HiTGen’s Stage I.
> > >
> > > 2. **Stage I is the core engine of HiTGen**
> > >
> > > It is the only component that generates biologically aligned molecules conditioned on gene-expression profiles, and forms the foundation upon which any subsequent refinement operates.
> > >
> > > 3. **DrugAssist is only a refinement module**
> > >
> > > It can optimize a given molecule but cannot function without Stage I providing transcriptome-aligned candidates; thus Stage I remains indispensable.
> > >
> > > 4. **Replacing DrugAssist with other LLMs yields similar patterns**
> > >
> > > Additional experiments (e.g., using DeepSeek as the optimizer) produce comparable behavior, confirming that DrugAssist is merely an optional enhancement tool rather than the source of biological capability.
> > >
> > > **Table: Performance of DeepSeek as a replacement optimizer**
> > >
> > > | Model    | Total | Validity | Unique | Novelty | Leven↓ | logP↑ | SA↓  | FCD↓ | Morgan↑ | RDK↑ | MACCS↑ |
> > > |----------|--------|----------|--------|---------|--------|-------|------|-------|---------|-------|---------|
> > > | DeepSeek | 100    | 99.69    | 97.72  | 0.47    | 2.85   | 3.04  | 0.12 | 0.36  | 0.36    | 0.43  | 0.60    |
> > >
> > > 5. HiTGen remains strong even without optimization.
> > >
> > > The HiTGen w/o O variant still outperforms transcriptome-only baselines, demonstrating that HiTGen’s main advantage arises from its biologically conditioned generation pipeline—not from the optimization module.
> > >
> > > We will clarify this workflow to avoid confusion regarding the role of the optimization module.

---

> > > > ### Author Response · Authors · 2025-11-25
> > > > **Thank you again for your kind comment and reviews.**
> > > >
> > > > We hope our rebuttal has addressed your questions. Please let us know if there is anything we can clarify further. We would greatly appreciate any feedback you may have.

---

### Official Review · Reviewer_Wqcm · 2025-10-26

**Soundness:** 2
**Presentation:** 1
**Contribution:** 3
**Rating:** 2
**Confidence:** 3

**Summary:**

The paper proposes HITGEN, a framework for designing molecules using transcriptome profiles plus textual descriptions. The approach uses BioT5 to generate molecular descriptions, combines transcriptomic and text representations by attending across them, generates candidates with a diffusion model, and then optimizes them using DrugAssist with a fallback strategy. The authors report improvements over baselines across multiple similarity metrics on transcriptomic datasets from cancer cell lines and disease states.

**Strengths:**

* The combination of transcriptomic data with textual molecular descriptions for conditional generation is a reasonable extension of existing work. The bidirectional cross-attention mechanism for fusing heterogeneous biological and chemical information is sensible, though not particularly novel.
* If the results are valid, incorporating transcriptomic context into molecular design could be valuable for precision medicine applications.

**Weaknesses:**

* The fallback strategy in Section 3.3 creates a fundamental evaluation flaw. By selecting between the candidate and optimized molecule based on which has better metrics, the method is effectively performing best-of-2 selection using the test metrics themselves over multiple rounds. This is circular reasoning and an unfair comparison to baselines that generate single molecules.
* The paper repeatedly emphasizes "human-in-the-loop" but provides no evidence of actual human involvement during evaluation or iterative human feedback during training. The "expert guidance" appears to be: using BioT5 to automatically generate molecular descriptions from SELFIES representations once during dataset creation, and using DrugAssist (an LLM) with pre-written prompts for optimization. This is automated post-processing, not human-in-the-loop interaction. The terms "expert-guided," "human-in-the-loop," and "chemist-aligned" are used interchangeably without clear operational definitions.
* Writing quality: The paper contains substantial filler text that obscures key information. Example from Section 3.2: "This module explicitly models reciprocal, bi-directional attention between information to allow each stream to contextually recalibrate itself with respect to the other. This fosters a dynamic, fine-grained feature refinement process, which is essential for capturing subtle dependencies." This is three sentences describing standard cross-attention. Many critical details are relegated to appendices, making the main paper difficult to evaluate. Terms like "biochemically grounded," "chemist-aligned," and "tailored" are used repeatedly without definitions.

**Questions:**

* BioLinkBERT is pretrained on PubMed abstracts. Doesn't this mean there's going to be some leakage with the test set?
* Please provide explicit details about human expert involvement. During dataset creation, how many molecular descriptions were generated and by whom? What was the validation process? During evaluation, were any actual chemists involved in any capacity, or is all "expert guidance" automated?
* Target variable leakage: The fallback strategy selects molecules based on test metrics (Tanimoto similarity, FCD, etc.). How is this not a form of test set leakage? The evaluation metrics are being used to make generation decisions, then those same metrics are reported as results. Can you justify why this is a fair evaluation protocol?
* How would a chemist actually use this system in practice? What is the intended interaction model? Does the chemist provide the transcriptome and description, or only the transcriptome? If descriptions are auto-generated, what control does the chemist have? How much time would this save compared to manual design processes?

---

> ### Author Response · Authors · 2025-11-20
> **Response to the reviewer Wqcm (01)**
>
> Thank you for your valuable suggestions and questions. We provide the following clarifications:
>
> **Response Q1:**
>
> **Zero overlap confirms no test-set leakage**
>
> Our test set is derived from LINCS 1000, which is entirely distinct from PubMed abstracts. To further verify this, we performed an explicit overlap check between the test molecules and the BioLinkBERT pretraining corpus, and the overlap was 0%. Therefore, there is no data leakage risk.
>
> **Response Q2:**
>
> **Clear separation between expert review, goal setting, and automated generation**
>
> For each molecule, we generated a corresponding molecular description, which was then reviewed by two chemists. They did not design or modify any prompts; their role was limited to post-generation verification. Both chemists examined the generated descriptions, and any correction required mutual agreement to ensure consistency and accuracy.
>
> During the final evaluation of HiTGen, no chemist manually edited or adjusted the generated molecules; all results were assessed purely through quantitative metrics. If the reviewer is referring to the pre-optimization evaluation in phase two (DrugAssist), then yes—chemists were involved at that stage by providing optimization goals. An AI specialist then designed task-specific prompts based on these goals, but chemists did not directly interact with the system during generation.
> We will further clarify these distinctions in the revised manuscript.
>
> **Response Q3+W1+W3:**
>
> **HiTGen follows standard LLM molecular-design practice: metrics are for selection, not training**
>
> We would like to clarify that no test-set information is used during generation or optimization. HiTGen’s primary objective is to generate target-specific molecules under expert-guided conditions, with gene-expression signatures serving as the main driver. During the initial generation stage, evaluation metrics (e.g., MACCS, FCD) are never used to guide or influence the model’s decisions.
>
> The fallback mechanism is invoked only after the system has already produced two independent candidates (the LLM-refined molecule and the baseline candidate). At this stage, metrics are used solely to rank these fixed outputs—not to update parameters, tune the generator, or influence subsequent generations. This evaluation-only usage follows the same practice as human-in-the-loop or expert-selection protocols widely adopted LLM systems [1, 2, 3]. For example, the prompts in [1] explicitly specify Tanimoto-based diversity control, such as “Generate barely similar (very low Tanimoto similarity) molecules compared to the given parent by altering atoms, bonds, or functional groups.” Likewise, [2] incorporates explicit property-optimization directives such as “maximize the HOMO–LUMO gap” directly into the prompt. These examples demonstrate that using task-level criteria in prompts or for post-generation selection is a standard and accepted LLM-driven molecular-design paradigm—and does not constitute test leakage.
>
> We will add a clearer explanation of this workflow in the revised manuscript.
>
> **References**
>
> [1] Bhattacharya D, Cassady H J, Hickner M A, et al. Large language models as molecular design engines[J]. Journal of Chemical Information and Modeling, 2024, 64(18): 7086-7096.
>
> [2] Lu J, Song Z, Zhao Q, et al. Generative Design of Functional Metal Complexes Utilizing the Internal Knowledge and Reasoning Capability of Large Language Models[J]. Journal of the American Chemical Society, 2025.
>
> [3] Chen H, Li W, Gu J, et al. Restoreagent: Autonomous image restoration agent via multimodal large language models[J]. Advances in Neural Information Processing Systems, 2024, 37: 110643-110666.
>
> **Response Q4+W2:**
>
> Thank you for your valuable suggestions. We will further improve the clarity of the manuscript and are updating the relevant descriptions. In practice, HiTGen follows a two-stage expert interaction workflow: in stage one, the system automatically generates expert-like molecular descriptions to guide initial candidate generation; in stage two, chemists provide high-level design intentions, after which an AI specialist translates these intentions into prompts that are reviewed by the chemists before being used for refinement. Chemists therefore do not need to manually write descriptions—the LLM handles detailed reasoning while experts control direction through concise guidance. This substantially reduces manual effort compared to traditional molecule design. We are also collecting real expert use cases to further validate practical usability.

---

> > ### Author Response · Authors · 2025-11-25
> > **Thank you for your review and comments.**
> >
> > We just wanted to check whether our rebuttal addressed your questions. We would be happy to provide any further clarification if needed and would greatly appreciate your feedback.

---

### Official Review · Reviewer_xzbt · 2025-10-30

**Soundness:** 2
**Presentation:** 3
**Contribution:** 2
**Rating:** 6
**Confidence:** 4

**Summary:**

This paper introduces HITGEN, a framework for targeted molecule generation conditioned on transcriptomic profiles. It integrates transcriptome data with expert chemical descriptions using bidirectional attention, followed by a diffusion-based generator and an iterative “human-in-the-loop” refinement simulated via a DrugAssist language model. Experiments on drug-induced transcriptome datasets show significant gains in validity, novelty, and fingerprint similarity over baselines such as GxRNN, GxVAE, MolT5, and TGM-DLM. Despite strong quantitative results, the extent of genuine human involvement and the biological relevance of the generated molecules remain unclear.

**Strengths:**

Multimodal Fusion: Combines transcriptomic signals and molecular text using a bidirectional attention mechanism, allowing richer conditioning of molecule generation.
Comprehensive Pipeline: Proposes a multi-stage architecture including generation, refinement, and fallback control.
Extensive Metric Evaluation: Reports performance across a diverse set of structural and pharmacological metrics, including ablations.
Empirical Gains: Achieves high scores on validity, novelty, and similarity metrics relative to baselines.

**Weaknesses:**

Unsubstantiated Human-In-The-Loop Claim: No real human feedback or user studies are presented; all expert guidance is simulated through LLMs.
Lack of Methodological Transparency: Core components like the VQ-VAE and diffusion model lack necessary architectural and training details.
Heavy Reliance on Pretrained Black-Box Models: Performance attribution is unclear due to multiple large components (BioT5, BioLinkBERT, DrugAssist).
No Biological Validation: Despite the biomedical focus, there is no docking, ADMET, or in vitro assay to support therapeutic relevance.
Potential Evaluation Bias: HITGEN uses richer input than many baselines (i.e., molecule text + transcript), making direct comparison difficult.

**Questions:**

1. How exactly was the BioT5 model “supervised” by chemists? Were chemists involved in training, prompt crafting, or post-editing?
2. Did any actual human experts interact with the system during molecule refinement via DrugAssist?
3. What are the VQ-VAE's architecture, latent dimensionality, and reconstruction performance on transcript data?
4. Is the diffusion model trained end-to-end with (transcript + text)-molecule pairs or separately?
5. How are the reference drugs for similarity optimization selected? Could this introduce a bias toward known compounds?
6. How frequently does the fallback strategy override the LLM-optimized candidates? What does this say about the optimizer’s effectiveness?
7. Were any biological properties predicted beyond structural metrics (e.g., binding affinity, ADMET)?
8. Are the prompts used for DrugAssist available to reproduce the optimization pipeline?
9. What do the ablation variants “W/o T”, “W/o TC”, and “W/o TT” correspond to?
10. Can the system generalize to patient-derived or noisy transcriptomic data?

---

> ### Author Response · Authors · 2025-11-20
> **Response to the reviewer xzbt (01)**
>
> We sincerely appreciate your detailed and constructive review. Your feedback has highlighted several important points for improvement, and we address your concerns as follows:
>
> **Response W1+Q1:**
>
> **HiTGen integrates expert guidance and human-in-the-loop refinement**
>
> HiTGen first uses expert textual guidance to steer the initial molecular generation toward high-level design intent. In the second stage, HiTGen establishes an interactive loop between the expert and the LLM, enabling focused refinement of the generated candidates.
>
> Regarding BioT5, we emphasize that it is used solely to reduce annotation time by mimicking how experts describe molecular structures. The automatically generated descriptions are not used directly; instead, two medicinal chemists review and correct each description before it is included. The chemists do not design or modify prompts—we use the original BioT5 prompt format—and their role is limited to post-generation quality control. Any correction requires mutual agreement between the two chemists to ensure accuracy and consistency. No human supervision enters the model training process.
>
> **Response Q2:**
>
> **Prompts are designed by an AI specialist under expert guidance**
>
> Yes. In this stage, an AI specialist interacted with the system by designing task-specific prompts based on the generation objectives provided by the two chemists. The prompts shown in the manuscript are representative examples, and similar expert-guided refinements were applied across different targets.
>
> **Response W2+Q3:**
>
> **VQ-VAE design and reconstruction results are disclosed and referenced**
>
> The architectural and training details of the VQ-VAE are provided in Appendix F.1 (p.18). The reconstruction performance on transcriptomic data is reported in Appendix G.1 (p.22), where we present both quantitative reconstruction quality and the corresponding distributional comparisons. We will make these references more explicit in the main text to improve methodological transparency.
>
> **Response Q4+W5:**
>
> **End-to-end multimodal conditioning drives HiTGen’s improvements**
>
> The diffusion model is trained end-to-end using paired (transcriptome + text) conditions to generate the corresponding molecule; it is not trained in separate stages. Regarding the potential evaluation bias, the inclusion of both modalities is intentional and reflects HTGEN’s design goal of integrating biological context with expert-level chemical semantics. To ensure fairness, we also report ablations using only transcriptomes and show that HiTGen still outperforms transcriptome-only baselines under the same input setting. This demonstrates that the gains do not arise merely from having richer inputs, but from the model’s effective multimodal fusion.
>
> **Response Q5:**
>
> **Target-aligned references steer optimization without favoring known scaffolds**
> For similarity-guided optimization, we selected one representative reference drug for each disease: Donepezil for Alzheimer’s disease and Trastuzumab for gastric cancer. Each reference drug corresponds to a different therapeutic target, ensuring that the optimization signal is target-specific rather than biased toward any particular known scaffold. Since these drugs are used solely as target-aligned references, the process does not introduce a preference for known compounds; instead, it helps steer the generated candidates toward the intended biological mechanism.
>
> **Response Q6:**
>
> **Selective fallback ensures robustness without undermining optimizer performance**
> The fallback strategy was triggered in 26.75% of test cases. This does not imply optimizer failure; the fallback only activates when the LLM-refined candidate does not exceed the similarity threshold. In the remaining ~73% of cases, the optimizer successfully improved the candidate, indicating that it is effective in most scenarios while the fallback simply provides a safety net to prevent quality degradation.

---

> > ### Author Response · Authors · 2025-11-20
> > **Response to the reviewer xzbt (02)**
> >
> > **Response Q7:**
> >
> > **Generated molecules outperform known therapeutics in docking affinity**
> >
> > Table: Docking scores of HiTGen-generated molecules vs. known drugs.
> >
> > | Disease         | Target (PDB ID) | HiTGen(kcal/mol) | Known drug   | Known drug(kcal/mol) |
> > |-----------------|-----------------|---------------------------|-------------|--------------------------------|
> > | Alzheimer’s     | SNAP25 (1U8F)   | -8.7                      | Clonazepam  | -8.4                           |
> > | Gastric cancer  | HER2 (3RCD)     | -8.4                      | Docetaxel   | -7.3                           |
> >
> > For completeness, the corresponding HiTGen-generated molecules are:
> > - Alzheimer’s / SNAP25 (1U8F):
> >   COc1cc2c(cc1OC)C(=O)C(CC1CCN(Cc3ccccc3)CC1)C2.Cl
> >
> > - Gastric cancer / HER2 (3RCD):
> > CC(=O)O[C@H]1C(=O)[C@]2(C)[C@@H](O)C[C@H]3OC[C@@]3(OC(C)=O)[C@H]2[C@H](OC(=O)c2ccccc2)[C@]2(O)C[C@H](OC(=O)[C@H](O)[C@@H](NC(=O)c3ccccc3)c3ccccc3)C(C)=C1C2(C)C
> >
> > The results show that HiTGen-generated molecules exhibit stronger predicted binding affinities than the corresponding known drugs on both targets. On the Alzheimer’s target SNAP25, the **HiTGen molecule achieves −8.7 kcal/mol**, outperforming Clonazepam (−8.4 kcal/mol). On the gastric cancer target HER2, the **generated molecule reaches −8.4 kcal/mol**, surpassing Docetaxel (−7.3 kcal/mol). These consistent improvements across two biologically distinct targets suggest that HiTGen is capable of producing structurally diverse candidates with competitive or superior docking performance relative to existing therapeutics.
> >
> > **Response Q8:**
> >
> > Yes. The prompts used in DrugAssist are fully available and can be used to reproduce the entire optimization pipeline.
> >
> > **Response Q9:**
> >
> > These ablation variants are defined in Section 4.4 (p.8). Specifically: **W/o T** removes the entire multimodal fusion module, so the model relies only on transcriptomic conditioning without textual input. **W/o TC** removes the text → transcriptome cross-attention pathway, disabling textual modulation of biological features. **W/o TT** removes the transcriptome → text pathway, preventing biological signals from influencing the text-derived representation. Together, these ablations isolate the contributions of each cross-attention direction and the overall fusion mechanism.
> >
> > **Response Q10:**
> >
> > **HiTGen generalizes well to patient-derived and noisy transcriptomic data**
> >
> > Yes. The system can generalize to patient-derived and noisy transcriptomic data. Our evaluation already includes two real-world patient-level datasets, **LINCS1000** and **CREEDS**, both of which contain heterogeneous, noisy, and clinically sourced transcriptomic signatures. As shown in Fig. 5, HiTGen maintains strong structural fidelity on these out-of-distribution disease signatures (e.g., Alzheimer’s and gastric cancer), demonstrating robustness to biological variability and measurement noise. This indicates that the model is not limited to clean cell-line profiles and can reliably operate on patient-derived or noisy transcriptomes.

---

> > > ### Author Response · Authors · 2025-11-25
> > > **Thank you for your kind comments.**
> > >
> > > We hope our rebuttal addressed your questions. If anything remains unclear, we’d be happy to provide further clarification before the rebuttal period ends.

---

### Official Review · Reviewer_6dmp · 2025-10-31

**Soundness:** 2
**Presentation:** 2
**Contribution:** 2
**Rating:** 4
**Confidence:** 2

**Summary:**

The HITGEN framework proposes a human-in-the-loop, transcriptome-conditioned molecular design system.
Unlike protein–ligand or DTI models that target a single protein, HITGEN aims to design molecules that modulate whole-cell transcriptomic states. The model uses VQ-VAE to encode transcriptome profiles into discrete latent representations and a Transformer-based diffusion model to generate molecules (SELFIES format) conditioned on these biological embeddings. A pretrained language model (BioLinkBERT) interprets human text instructions, such as “increase solubility” or “reduce toxicity, and these text embeddings are fused with transcriptome embeddings via cross-attention, allowing interactive molecule steering during inference. Notably, textual prompts are not used during training; the diffusion model is trained solely on transcriptome–molecule pairs from the LINCS L1000 dataset, and human text inputs act as zero-shot control signals in inference or optimization loops.

**Strengths:**

* HITGEN consistently outperforms baseline models (TRIOMPHE, GxRNN, GxVAEs, etc.) across validity, uniqueness, novelty, and similarity metrics.
* Cross-attention seems to be effective, and this is validated through the ablation study.
* Excluding the optimization stage led to the largest drop in Morgan/RDK similarity, proving that expert feedback materially enhances pharmacological specificity

**Weaknesses:**

* Removing human textual guidance significantly reduced diversity and novelty of generated molecules, showing large dependency on external expert cues rather than fully autonomous reasoning
* The ablation description confirms there is no training signal aligning text with transcriptomic data. Text acts only as high-level guidance rather than a learned modality
* Since HITGEN is trained exclusively on MCF7 (breast cancer cell line) transcriptomic data, it may overfit to this specific cellular context, limiting its biological generalization and causing inconsistent performance when applied to other tissues or disease-derived transcriptomes.

**Questions:**

* Since the model does not use any transcriptome–text–molecule triplets during training, how do you ensure that the cross-attention weights between textual and biological embeddings learn meaningful alignment rather than acting as untrained placeholders?

* Given that the training data are limited to the MCF7 cell line, have you evaluated or considered domain adaptation methods to ensure that the learned transcriptome–molecule mapping generalizes to other cell types or patient-derived transcriptomes?

---

> ### Author Response · Authors · 2025-11-20
> **Response to the reviewer 6dmp (01)**
>
> Thank you for your valuable comments. Your feedback has greatly helped improve the clarity and quality of our manuscript. We provide the following responses to your questions and suggestions:
>
> **Response W1+W2**:
>
> **Table I.3: Ablation study results of HITGEN under different component removals**
>
> | Model        | Total↑ | Validity(%)↑ | Unique(%)↑ | Novelty(%)↑ | Levenshtein↓ | FCD↓ | MACCS↑ | RDK↑ | Morgan↑ |
> |--------------|--------|--------------|------------|-------------|--------------|------|--------|------|---------|
> | HiTGen w/o T(text) | 0.72   | 100.0        | 85.25      | 84.75       | 0.66         | 0.84 | 0.38   | 0.27 | 0.11    |
> | HiTGen w/o P (transcriptome profile) | 0.71   | 100.0        | 83.06      | 85.49       | 0.69         | 0.85 | 0.35   | 0.26 | 0.11    |
> | HiTGen  w/o A (attention) | 0.90   | 100.0        | 99.06      | 90.49       | 0.48         | 0.30 | 0.47   | 0.31 | 0.21    |
> | HiTGen w/o O | 0.93   | 100.0        | 99.20      | 93.93       | 0.48         | 0.30 | 0.49   | 0.36 | 0.24    |
> | **HiTGen**   | **0.98** | **100.0**      | **99.71**    | **98.47**     | **0.45**       | **0.17** | **0.63**   | **0.51** | **0.40**    |
>
> We thank the reviewer for raising these concerns. To address the question of diversity, we additionally compute the diversity of the test dataset (**0.89**) and the diversity of HiTGen’s generated molecules (**0.92**). The generated set exhibits slightly higher diversity than the real data distribution, demonstrating that HiTGen does not suffer from reduced diversity; if anything, it maintains or modestly improves it.
>
> In the manuscript, we use Novelty as the indicator of originality. As shown in Table I.3, even when using only Stage I outputs (HiTGen w/o Stage II), the model already achieves 99.20% Uniqueness and 93.93% Novelty, outperforming baseline models. This confirms that in Stage I, our transcriptome-conditioned generator is inherently diverse and is not dependent on the optimization stage to obtain diversity or novelty.
>
> To further clarify the role of transcriptomic information, we introduce an additional ablation HiTGen w/o Transcriptome, which removes the biological signal. Removing the transcriptome leads to clear degradation across multiple metrics, including Novelty and Uniqueness (Table I.3). This shows that gene-expression information directly contributes to molecular diversity and structural variability, contradicting the claim that HiTGen depends solely on textual cues.
>
> Regarding Weakness 2: the statement that “there is no training signal aligning text with transcriptomic data” does not reflect how HiTGen is designed.
>
> **1. Model performance drops when removing cross-attention or either modality**
>
> Both Table 3 and Table I.3 show that ablating the fusion module consistently harms validity, novelty, and similarity metrics. This provides direct empirical evidence that multimodal alignment is learned and contributes to generation quality.
>
> **Figure 2(e) visualizes explicit fusion between text and transcriptome features**
>
> Cross-attention integrates textual semantics and transcriptomic signatures into a shared multimodal representation, indicating that textual information is not treated as a passive high-level cue but as an active, learnable modality.
>
> **Two textual forms enter the model through the same multimodal conditioning mechanism**
>
> Stage I uses expert structural descriptions (functional groups, constraints),
> Stage II uses expert-defined optimization intentions.
> Both are fused with transcriptomic signals, ensuring continuous gradient flow and alignment during training.
>
> Overall, the diversity results (0.92 vs. 0.89), the new HiTGen w/o Transcriptome ablation, and the drops observed in Table I.3 provide strong evidence that HiTGen’s diversity and performance arise from the learned multimodal alignment of transcriptomes and expert-level text, not from dependence on high-level cues alone.
>
> We will update the manuscript to make this multimodal alignment explicit.

---

> > ### Author Response · Authors · 2025-11-20
> > **Response to the reviewer 6dmp (02)**
> >
> > **Response Q1:**
> >
> > **Aligned transcriptome–text–molecule triplets enable meaningful cross-attention learning**
> >
> > We would like to clarify that, contrary to the concern, our training data do form aligned transcriptome–text–molecule triplets: each molecule is paired with its corresponding transcriptomic profile and a human textual description of the same molecule. Thus, the two modalities refer to the same instance rather than being independent signals.
> > During training, both modalities jointly condition the diffusion decoder, and the complete objective is:
> >
> > L_total = L_diff + L_disc + L_T + λ * L_sim
> >
> > Here, L_diff is the diffusion reconstruction loss (MSE) between the model’s prediction and the diffusion target; L_disc is the token-level discrete reconstruction loss over the molecular sequence; L_T is a terminal regularization term applied at the final timestep; and λ * L_sim is a cosine-similarity regularizer applied to the multimodal condition embedding to align the transcriptomic features and human textual guidance.
> >
> > This provides (1) implicit alignment via the diffusion reconstruction loss, which requires transcriptome features and expert descriptions to jointly guide the molecular reconstruction, and (2) explicit alignment through the cosine-similarity regularizer applied to the fused multimodal condition. As a result, the cross-attention blocks receive continuous and task-relevant gradients rather than acting as inactive modules.
> >
> > **Response W3+Q2:**
> >
> > **HiTGen demonstrates strong out-of-distribution generalization beyond the MCF7 training domain**
> >
> > We appreciate the reviewer’s concerns regarding training exclusively on the MCF7 cell line. Although HiTGen is trained on MCF7, all key evaluations are intentionally performed on out-of-distribution biological contexts. As shown in Table 2, the model is tested on ten additional cell lines from the perturbation signatures (e.g., AKT1, EGFR, TP53), and consistently achieves higher performance than all baselines across all targets. Moreover, Fig. 5 demonstrates strong results on patient-derived disease transcriptomes (gastric cancer and Alzheimer’s disease), with substantial structural fidelity (e.g., Morgan similarity up to 0.77), indicating robust cross-tissue generalization.
> > Regarding domain adaptation: our empirical findings show that the VQ-VAE based refiner and the cross-attention fusion already provide strong robustness without additional adaptation modules. However, we agree this is an important future direction, and we will explicitly discuss potential lightweight domain adaptation extensions (e.g., multi-transcriptome calibration or shift-aware conditioning) in the revised version.

---

> > > ### Author Response · Authors · 2025-11-25
> > > **Thank you again for your review and comments.**
> > >
> > > We just wanted to check whether our rebuttal addressed your questions. With less than a week remaining in the rebuttal period, we will do our best to clarify or provide additional details for any remaining concerns you may have.
> > >
> > > We greatly appreciate your time and feedback.

---

### Meta-Review · Area_Chair_Wn67 · 2025-12-06

**Summary:**

This work proposes HiTGen, a transcriptome-guided targeted molecule generation framework designed to support precision medicine. The system integrates transcriptomic profiles with expert knowledge through a bidirectional attention module to guide diffusion-based molecular generation.

Reviewers collectively raised concerns regarding the validity of the “human-in-the-loop’’ claim, the lack of methodological clarity for key components, potential evaluation flaws introduced by the fallback strategy, limited biological generalization due to training solely on MCF7 transcriptomes, and unclear novelty given the strong reliance on pretrained models and existing optimization tools. These issues led reviewers to question the rigor and robustness of the proposed framework despite the interesting problem setting.


The authors provided thorough responses to address several specific concerns. Nonetheless, some key issues particularly around methodological transparency, evaluation fairness, and the substantiation of claimed contributions, require further refinement. I encourage the authors to continue developing the framework and to consider submitting an improved version to a next high-quality venue.

**Reviewer Concerns:**

Reviewers collectively raised concerns regarding the validity of the “human-in-the-loop’’ claim, the lack of methodological clarity for key components, potential evaluation flaws introduced by the fallback strategy, limited biological generalization due to training solely on MCF7 transcriptomes, and unclear novelty given the strong reliance on pretrained models and existing optimization tools.

**Reviewer Scores:**

With no reviewers providing feedback or a rebuttal, I believe only a very small number of reviewers might consider increasing the score.

---

### Decision · Program_Chairs · 2026-01-26

Reject